# Unsupervised Domain Adaptation via Minimized Joint Error

**Dexuan Zhang**                                                    *dexuan.zhang@mi.t.u-tokyo.ac.jp*
*The University of Tokyo*

**Thomas Westfechtel**                                             *thomas@mi.t.u-tokyo.ac.jp*
*The University of Tokyo*

**Tatsuya Harada**                                                 *harada@mi.t.u-tokyo.ac.jp*
*The University of Tokyo*
*RIKEN*

**Reviewed on OpenReview:** *https://openreview.net/forum?id=kiPsMct7vL&noteId=r5dySaCR1o*

## Abstract

Unsupervised domain adaptation transfers knowledge from a fully labeled source domain to a different target domain, where no labeled data are available. Some researchers have proposed upper bounds for the target error when transferring knowledge. For example, Ben-David et al. (2010) established a theory based on minimizing the source error and distance between marginal distributions simultaneously. However, in most research, the joint error is ignored because of its intractability. In this research, we argue that joint errors are essential for domain adaptation problems, particularly when the domain gap is large. To address this problem, we propose a novel objective related to the upper bound of the joint error. Moreover, we adopt a source/pseudo-target label-induced hypothesis space that can reduce the search space to further tighten this bound. To measure the dissimilarity between hypotheses, we define a novel cross-margin discrepancy to alleviate instability during adversarial learning. In addition, we present extensive empirical evidence showing that the proposed method boosts the performance of image classification accuracy on standard domain adaptation benchmarks.

## 1 Introduction

Traditional machine-learning theories generally assume that training and test data are drawn from an identical distribution, that is, the same domain. However, this assumption does not necessarily hold in real-world settings. Considering image classification as an example, several factors, such as the change in light, noise, angle at which the image is captured, and different types of sensors, can lead to a domain gap that harms the performance when predicting the test data. In many practical cases, a model trained in one domain is expected to be applied to another. As a solution, domain adaptation (DA) aims to transfer the knowledge learned from a source domain, which is typically fully labeled, into a different (although related) target domain. This research focused on the most challenging case, unsupervised domain adaptation (UDA), in which no target label is available.

Ben-David et al. (2010) suggested that the target error can be minimized by bounding the error on source data, the discrepancy between domains, and a small optimal joint error. The optimal joint error is quantified by the lowest error rate that a hypothesis can achieve in both domains. Many researchers have focused on learning domain-invariant features such that the discrepancy between domains can be minimized. Two strategies were explored for aligning the domains. The first method bridges domains by matching their statistics (Long et al., 2015; 2017; Pan et al., 2009). The second method utilizes adversarial learning (Goodfellow et al., 2014)

to develop a minimax game in which a domain discriminator is trained to distinguish the source from the target domain while the feature extractor is trained to confuse it simultaneously (Ganin & Lempitsky, 2015; Ganin et al., 2016; Tzeng et al., 2017). Despite the remarkable performance achieved by domain-matching schemes, they still suffer from a major limitation: the joint distributions of feature spaces and categories are not well-aligned across domains. As reported in Ganin et al. (2016), such methods fail to generalize to closely related source–target pairs, for example, the adaptation from MNIST (LeCun et al., 1998) to SVHN (Netzer et al., 2011). One potential reason is that when matching the marginal distributions of the two domains, samples from different classes can be mixed up, where the joint error becomes nonnegligible because no hypothesis can jointly classify the source and target data with high accuracy.

This research aims to address the aforementioned problem by incorporating the joint error to formalize an optimizable upper bound such that the mismatch of the joint distributions can be properly penalized. Adversarial learning is effective for aligning distributions (Ganin et al., 2016; Tzeng et al., 2017; Saito et al., 2017b; Zhang et al., 2019b); however, it can suffer from instability owing to rapid changes in the discriminator. We propose a specially designed distance measurement based on margin theory that has a smoother gradient around the decision boundary to alleviate this instability, namely, cross-margin discrepancy (CMD). In addition, when we apply CMD to the proposed target error bound, we can prove that part of the objective can be transformed into a CGAN (Mirza & Osindero, 2014) objective [1], which is effective in aligning conditional distributions. We evaluate the proposed method using several classification tasks. Our method improves performance by a large margin, particularly when the domain gap is large. The contributions of this research can be summarized as follows:

- We propose a novel objective that relates to an upper bound of the joint error and show that our proposal can reduce the chance of misalignment through the distribution matching.

- We create a specific hypothesis space induced by source/pseudo-target labels to stiffen the proposed objective and avoid optimizing a loose bound within an immense searching space.

- We adopt a novel measurement, namely CMD, which measures the dissimilarity between hypotheses. We demonstrate that using this measurement, we can alleviate the instability during adversarial learning.

- We provide extensive empirical evidence showing that our proposal outperforms other upper bound related methods in image classification on several DA benchmarks, particularly when the domain gap is large.

## 2 Related Work

The upper bound proposed by Ben-David et al. (2010) invoked numerous approaches that focused on reducing the discrepancy between the source and target domains by learning domain-invariant features. Long et al. (2015; 2017) utilized maximum mean discrepancy (MMD) to match the hidden representations of certain layers in a deep neural network. Pan et al. (2011) proposed transfer component analysis to learn a subspace across domains in a reproducing kernel Hilbert space using MMD, which dramatically minimizes the discrepancy between domains. Li et al. (2018) applied adaptive batch normalization to modulate statistics from the source to the target domain in the batch normalization layers across the network in a parameter-free manner.

Another way to learn domain-invariant features is to leverage generative adversarial networks. Ganin & Lempitsky (2015) relaxed the discrepancy measurement in the upper bound from Ben-David et al. (2010), where a domain classifier is trained to distinguish different domains while a feature extractor is trained to confuse it. Tzeng et al. (2017) followed this idea but divided the training procedure into a classification stage and an adversarial learning stage. Saito et al. (2017b) explored a tighter bound by explicitly utilizing task-specific classifiers as discriminators such that features near the support of source samples will be favored by the extractor. Zhang et al. (2019b) introduced margin disparity discrepancy, a novel measurement with rigorous generalization bounds, to bridge the gap between the theory and algorithm. Methods that perform

---

[1]In Eq.9, we show that our proposal includes a CGAN objective that aligns three pairs of hypotheses induced distributions

distribution alignment at the pixel level in raw images, known as image-to-image translation, have been explored by Liu & Tuzel (2016); Bousmalis et al. (2017); Sankaranarayanan et al. (2017); Shrivastava et al. (2016); Hoffman et al. (2018); Murez et al. (2017).

Distribution matching can not only bring domains closer, but also mix samples from different classes. Therefore, Sener et al. (2016); Zhang et al. (2018) used pseudo-labels to learn discriminative target representations. However, this typically requires a complex labeling procedure and data-dependent hyperparameters to set a threshold for a reliable prediction. Long et al. (2018) proposed a framework that conditions adversarial adaptation models on the discriminative information conveyed in classifier predictions. However, it matches the marginal distributions of joint distributions for features and pseudo-predictions, which require a balanced label assumption to achieve the alignment of conditional distributions. Wu et al. (2019) raised attention to the increasing joint error caused by distribution matching and proposed a relaxed match by restricting the power of domain classifier to handle the label shifting problem. However, this requires an overlap between the source and target domains in the input space and does not necessarily reduce the joint error. Saito et al. (2017a) claimed that their proposal can reduce the joint error along with the discrepancy between domains by progressively creating a pseudo-labeled target set. However, it reforms the joint error as the sum of the error on the source and pseudo-labeled target domains and a false label rate that cannot be optimized. In this research, we proposed a theoretical approach that constructs an optimizable upper bound on the joint error for UDA tasks.

In summary, the upper bound proposed by Ben-David et al. (2010) and its extensions (Ganin et al., 2016; Tzeng et al., 2017; Saito et al., 2017b; Zhang et al., 2019b) continue to improve the alignment of the marginal distributions of the source and target domains. However, the problem of ignoring joint errors remains unsolved. In the following sections, we theoretically explain the ways by which it can harm performance in DA tasks and the ways our proposal can handle this problem.

## 3 Proposed Method

In this section, we present details of the proposed method. First, in Sec.3.1, we propose a target error bound and show its relation to the joint error. Then, we theoretically explain the importance of joint errors in DA. Second, in Sec.3.2, we introduce approximated labeling functions inside constrained hypothesis space to formalize an objective that can be optimized. Finally, in Sec.3.3, we propose a novel measurement for the dissimilarity between hypotheses based on the margin theory and show its utility in adversarial learning.

### 3.1 Upper Bound Incorporating Joint Error

We consider the UDA as a multiclass classification task in which the learning algorithm has access to a set of $n$ labeled points $\{(x_s^i, y_s^i) \in (X \in \mathbb{R}^D \times Y = \{1, ..., K\})\}_{i=1}^n$ sampled i.i.d. from the source domain $S$ and set of $m$ unlabeled points $\{(x_t^i) \in X \in \mathbb{R}^d\}_{i=1}^m$ sampled i.i.d. from target domain $T$. Let $f_S : X \in \mathbb{R}^D \to \mathbb{R}^K$ and $f_T : X \in \mathbb{R}^D \to \mathbb{R}^K$ be the true labeling functions in the source and target domains, respectively, and their outputs are one-hot vectors denoting the corresponding classes of inputs. Let $\epsilon_D(f, f')$ denote a distance metric that measures the expectation of disagreement between two functions $f, f'$ over distribution $D$. To refer to the source error of a hypothesis $h \in H : X \in \mathbb{R}^D \to \mathbb{R}^K$, we use the shorthand $\epsilon_S(h) := \epsilon_S(h, f_S)$, which measures the disagreement w.r.t. the true labeling function $f_S$ over the domain $S$. Similarly, we use $\epsilon_T(h)$ to denote the target error (the second argument of $\epsilon$ is omitted when we refer to the source and target errors according to Ben-David et al. (2010)). **With these notations, we propose the following upper bound for target error**[2]:

$$\epsilon_T(h) \leq \epsilon_S(h) + \epsilon_T(f_S, f_T) + \epsilon_S(f_S, f_T) + \epsilon_T(h, f_S) - \epsilon_S(h, f_T)$$
$$= \epsilon_S(h) + C_{S,T}(f_S, f_T, h) \tag{1}$$

For simplicity, we use $C_{S,T}(f_S, f_T, h)$ to denote $\epsilon_T(f_S, f_T) + \epsilon_S(f_S, f_T) + \epsilon_T(h, f_S) - \epsilon_S(h, f_T)$, which indicates a discrepancy between the domains. The aforementioned upper bound is minimized when $h = f_S$, and the

---

[2]See proof in A.6

minimum is equal to $\epsilon_T(f_S, f_T)$ given the triangle inequality[2]. Furthermore, we demonstrate that, in such a case, **our proposal becomes the upper bound of the optimal joint error[2] $\lambda$.**

**The reason we focus on the joint error is that** Ben-David et al. (2010) state the target error can be upper bounded by the sum of the source error and the marginal discrepancy between domains and the joint error:

$$\epsilon_T(h) \leq \epsilon_S(h) + \max_{f_1, f_2 \in H} |\epsilon_S(f_1, f_2) - \epsilon_T(f_1, f_2)| + \lambda$$

However, many works (Ganin et al., 2016; Tzeng et al., 2017; Saito et al., 2017b; Zhang et al., 2019b; Kim et al., 2019) choose to ignore $\lambda$ and mainly focus on minimizing the marginal discrepancy within a transformed input space $S_g = \{(g(x_s), y_s) | (x_s, y_s) \sim S\}, T_g = \{g(x_t) | x_t \sim T\}$ introduced by a feature extractor $g : X \in \mathbb{R}^D \to \mathbb{R}^F$ and $h, f_1, f_2 \in H^F : \mathbb{R}^F \to \mathbb{R}^K$:

$$\min_{h \in H^F, g} [\epsilon_{S_g}(h) + \max_{f_1, f_2 \in H^F} |\epsilon_{S_g}(f_1, f_2) - \epsilon_{T_g}(f_1, f_2)|] + \lambda_g,$$

where joint error is $\lambda_g = \min_{h^* \in H^F} \epsilon_{S_g}(h^*) + \epsilon_{T_g}(h^*)$. Without the feature extractor, the joint error is an independent term. **However, when we introduce the feature extractor $g$ to minimize the marginal discrepancy, we also violate the independence. The joint error varies with the change in $g$ and can become nonnegligible if samples from different classes are mixed, particularly when a large domain gap exists** (Zhao et al., 2019). In this case, regardless of the way we minimize the marginal discrepancy, the target error is unbounded because not a single $h$ can jointly classify both domains.

In Fig. 1b, we illustrate a case in which common methods fail to penalize mismatches. Without supervision, samples from different classes can be aligned during marginal distribution matching, which introduces large joint errors. This can be observed on the right-hand side of Fig. 1b, whereas the marginal discrepancy (areas 1, 3, 4, and 6) is relatively small, a large joint error (areas 2 and 5) exists. In this case, reducing the marginal discrepancy (areas 1, 3, 4, and 6) increases the joint error (areas 2 and 5). Because the two areas represent the overlaps of different classes, no hypothesis can correctly classify the samples in these areas, leading to a nonnegligible joint error. In the case illustrated in Fig. 1b, $f_S$ is assumed to have a specific shape for simplicity. Then, $\epsilon_T(f_S, f_T)$ (the minimum of our upper bound) exactly measures the overlapping areas 2 and 5, which are equivalent to the optimal joint error. Our proposal is closely related to the optimal joint error; thus, it can adequately reduce the size of the incorrect overlapping during distribution matching. In addition, according to Mansour et al. (2009), we provide the Rademacher complexity bound in A.3.

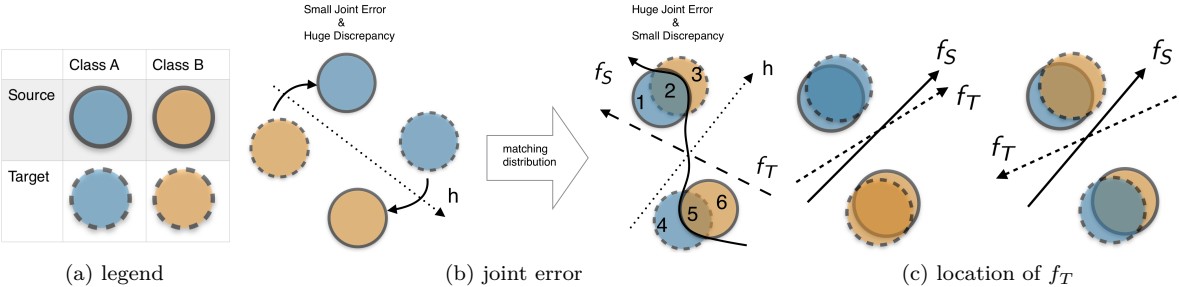

(a) legend        (b) joint error        (c) location of $f_T$

Figure 1: Left side of a classifier (arrow) is class A and the right side is class B; (a) Legend; (b) Joint error (areas 2 and 5) is unbounded by a simple distribution matching; (c) even with the marginal distribution aligned, $f_S$ and $f_T$ may be significantly different.

## 3.2 Hypothesis Space Constraint

Because the true labeling functions $f_S, f_T$ are not available, **we relax the upper bound Eq.1 by considering maximum** w.r.t. approximated labeling functions $f_1, f_2$ within a hypothesis space $H$[3]:

$$\epsilon_T(h) \leq \epsilon_S(h) + C_{S,T}(f_S, f_T, h) \leq \epsilon_S(h) + \max_{f_1, f_2 \in H} C_{S,T}(f_1, f_2, h)$$

---

[3]Our proposal holds even if $f_S, f_T \notin H$, which is proved in A.4

However, if we leave $H$ unconstrained, the maximum term can be arbitrarily large. To obtain a tight bound, the size of the hypothesis space must be restricted and an upper bound must be maintained. **We assume that it is possible to create two subspaces $H_1, H_2 \subseteq H$ such that the following holds**:

$$C_{S,T}(f_S, f_T, h) \leq \max_{f_1 \in H_1, f_2 \in H_2} C_{S,T}(f_1, f_2, h) \leq \max_{f_1, f_2 \in H} C_{S,T}(f_1, f_2, h) \tag{2}$$

The right-hand side of the aforementioned inequality always holds because of the subspace relationship. A sufficient condition for the left-hand inequality is $f_S \in H_1, f_T \in H_2$. The condition for $H_1$ can be easily achieved because we have source labels to create a space consisting of all the classifiers for the source domain, namely $H_{sc}$ that must contain $f_S$. However, the condition for $H_2$ is slightly problematic because we have no access to the true labels of the target domain; therefore, locating $f_T$ is difficult. Consequently, we construct a constrained hypothesis space for $H_2$ that is likely to contain $f_T$. **Generally, it is difficult to demonstrate the validity of the left side inequality of Eq.2 theoretically. Alternatively, we present the experimental results of the real-world problem to support our assumption**[4].

As illustrated in Fig. 1c, if the conditional distributions are well-aligned (for example, the left-hand side of Fig. 1c), after matching the source and target domains, it is reasonable to assume that $f_T \in H_{sc}$. However, good alignment cannot always be achieved, particularly when the domain gap is large (for example, the right-hand side of Fig. 1c). Considering this, **in the following sections, we propose a hypothesis space based on the constraints for $H_2$**, which aims to alleviate the worst case caused by the unknown location of $f_T$.

### 3.2.1 Target-driven Hypothesis Space (THS)

We first create a space in which the hypothesis can classify all the samples from the source domain, namely $H_{sc}$. Then, we create a space $H_{\tilde{t}c}$ consisting of all classifiers for the pseudo-labeled target domain $\{(x_t^i, \tilde{y}_t^i)) \in X \times Y\}_{i=1}^m$ ($\tilde{y}_t^i$ is provided by the prediction of $h$), where components $f \in H_{\tilde{t}c}$ can minimize $\tilde{\epsilon}_T(f) := \epsilon_T(f, h)$. **We further define $H_2$ as an intersection between two hypothesis spaces, i.e., $H_{sc}^{\gamma} \cap H_{\tilde{t}c}^{1-\eta}$**, where the hypothesis can classify samples from the source domain with an accuracy of $\gamma \in [0,1]$ and classify samples from the approximated target domain with an accuracy of $1 - \eta \in [0,1]$ (**generally, we set $\gamma = \eta$ for simplicity; however, in some experiments we tune the two parameters to show the influence when restricting the belief toward the source predictions and leveraging the pseudo-predictions**). In practice, it is difficult to create such a space and sample from it owing to its high computational cost. Alternatively, we use a weighted average to constrain the behavior of $f_2$ as an approximation of the sample from $H_{sc}^{\gamma} \cap H_{\tilde{t}c}^{1-\eta}$, which enables us to use $\eta$ to balance the confidence in the source and pseudo-target domains. This leads to the following constraints:

$$\begin{cases} H_1 = \{f_1 | \arg\min_{f_1 \in H}[\epsilon_S(f_1)]\} \\ H_2 = \{f_2 | \arg\min_{f_2 \in H}[\gamma \epsilon_S(f_2) + (1-\eta)\tilde{\epsilon}_T(f_2)]\} \end{cases} \tag{3}$$

### 3.2.2 Intuition to the Difference between Hypothesis Spaces

The reason for placing the aforementioned constraints on the hypothesis space $H_2$ can be intuitively explained by Fig. 2. If we set the hypothesis space $H_2$ to $H_{sc}$ (a set of classifiers for all source samples), classifier $f_2 \in H_2$ is forced to correctly classify all source samples. This increases the probability that all target samples are outside the decision boundary of $f_2$, particularly when the domain gap is large. In this case, these samples can be moved to either side of the decision boundary of $f_2$ to reduce objective $\epsilon_T(f_1, f_2)$ (shaded area in Fig. 2a), which may cause incorrect alignment. By constructing $H_2$ as an intersection of two hypothesis spaces $H_{sc}^{\gamma}$ (a set of classifiers for some of the source samples), $H_{\tilde{t}c}^{1-\eta}$ (a set of classifiers for some of the pseudo-labeled target samples) in Fig. 2b, we alleviate the restrictions on $f_2$ such that the decision boundary can pass through at least a few target samples where the objective (shaded area) can be properly reduced by

---

[4]We show that our proposal remains a valid upper bound, even if the domain gap is as large as we are unlikely to construct a hypothesis space that contains $f_T$ in A.5

moving the samples inside the decision boundary of $f_2$. In addition, given these hypothesis space constraints, **we demonstrate that under certain conditions, our proposal reduces to other UDA methods**[5].

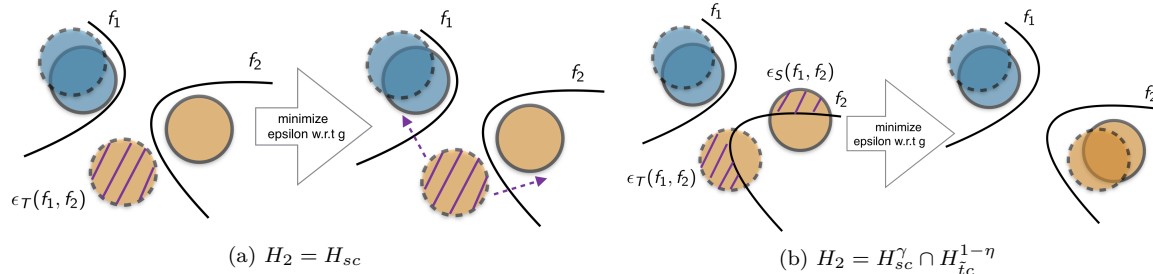

(a) $H_2 = H_{sc}$          (b) $H_2 = H_{sc}^{\gamma} \cap H_{\hat{t}c}^{1-\eta}$

Figure 2: Improper hypothesis space constraint can cause wrong alignment. (a) If $f_2$ is forced to only classify all source samples, it may completely misclassify all target samples when the domain gap is large; (b) If $f_2$ can classify a part of source samples and pseudo-labeled target samples, $S, T$ can be aligned in a desired way by minimizing the upper bound (shadow area).

### 3.3 Cross-Margin Discrepancy (CMD)

In this section, inspired by the margin theory, **we propose a novel discrepancy measurement called the CMD for $\epsilon$** instead of the commonly used loss functions (e.g., logistic, hinge, $L_1$) to stabilize the training of adversarial learning. Following the aforementioned notation, we define the measurement of the discrepancy between any two hypotheses $f_1, f_2 \in H : X \in \mathbb{R}^D \to \mathbb{R}^K$ (e.g., a multilayer perceptron with an output layer of a softmax function) over a distribution $D$:

$$\epsilon_D(f_1, f_2) = \mathbb{E}_{x \in D}[\mathbf{cmd}(f_1, f_2; x)] \tag{4}$$

We then consider the probability function $f(y|x)$, which indicates the y-th element of the output of $f \in H : X \in \mathbb{R}^D \to \mathbb{R}^K$ evaluated on input $x$. Thus, an induced labeling function, named $l_f$ from $X \to Y$ is expressed by

$$l_f : x \to \arg\max_{y \in Y} f(y|x)$$

**When two hypotheses $(f_1, f_2)$ disagree on $x$**, i.e., $y_1 = l_{f_1}(x) \neq l_{f_2}(x) = y_2$, the loss is defined as follows:

$$\mathbf{cmd}(f_1, f_2; x) = \log f_1(y_1|x) + \log(1 - f_1(y_2|x)) + \log f_2(y_2|x) + \log(1 - f_2(y_1|x)) \tag{5}$$

**When two hypotheses $(f_1, f_2)$ agree on $x$**, i.e., $y = l_{f_1}(x) = l_{f_2}(x)$, the loss is defined as follows:

$$\mathbf{cmd}(f_1, f_2; x) = \log \max(f_1(y|x), f_2(y|x)) + \log \max(1 - f_1(y|x), 1 - f_2(y|x)) \tag{6}$$

**We demonstrate that CMD can be viewed as: a sum of two well-established margin losses**[6], which play a significant role in achieving strong generalization performance (Koltchinskii & Panchenko, 2002); **an objective of CGAN**[7], which is an efficient method to align conditional distributions (Mirza & Osindero, 2014). Another reason to propose such a discrepancy measurement is that it  textbf helps alleviate instability in adversarial learning. As illustrated in Fig. 3b, during the optimization of a minimax game, when two hypotheses attempt to maximize the discrepancy (shaded area), if one hypothesis moves exceedingly fast around the decision boundary such that the discrepancy is maximized w.r.t. some samples, then these

---

[5]See the proof in A.2

[6]See proof in A.8

[7]See proof in A.9

samples can be moved to either side to decrease the discrepancy by tuning the feature extractor. As shown in Fig. 3a, the CMD is flat around the original, that is, the gradient with respect to the points near the decision boundary is relatively small. This can help prevent the aforementioned failure because each update of the hypotheses will be subtle during training.

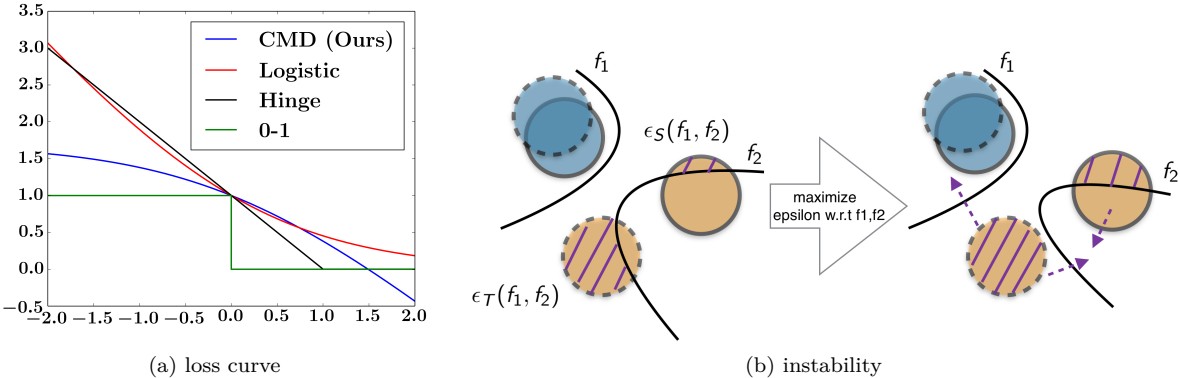

(a) loss curve                    (b) instability

Figure 3: (a) Losses of binary classification, where the proposed method shows small gradient around the original; (b) Steep gradients near the original may lead to an extreme move of the decision boundary.

### 3.4 Training Procedure

We introduce a feature extractor $g : X \in \mathbb{R}^D \to \mathbb{R}^F$ and the transformed feature space $S_g = \{(g(x_s), y_s)|(x_s, y_s) \sim S\}, T_g = \{g(x_t)|x_t \sim T\}$ as well as hypotheses $h, f_1, f_2 \in H^F : \mathbb{R}^F \to \mathbb{R}^K$.

**First, we define the source error of $h$ based on CMD:**

$$
\begin{aligned}
\boldsymbol{L_{ce}} = \epsilon_{S_g}(h) = \epsilon_{S_g}(h, f_S) &= \mathbb{E}_{x_s, y_s \in S}[\mathrm{cmd}(h, f_S; g(x_s))] \\
&= \mathbb{E}_{x_s, y_s \in S}[\log f_S(y_s|x_s) + \log(1 - h(y_s|g(x_s)))] \\
&\approx -\mathbb{E}_{x_s, y_s \in S} \log h(y_s|g(x_s)),
\end{aligned}
$$

where the source error can be expressed as a cross-entropy loss because the true labeling functions map the inputs to one-hot vectors, denoting their corresponding labels. The source errors of $f_1, f_2$ denoted by $\epsilon_{S_g}(f_1), \epsilon_{S_g}(f_2)$ can be derived analogously.

**Second, we define constraints for approximated labeling functions $f_1, f_2$ to ensure they exist in the proper hypothesis spaces $H_1, H_2$ based on Eq.3:**

$$
\begin{cases}
\boldsymbol{L_{H_1}} = \epsilon_{S_g}(f_1) \\
\boldsymbol{L_{H_2}} = \gamma \epsilon_{S_g}(f_2) + (1 - \eta)\tilde{\epsilon}_{T_g}(f_2),
\end{cases}
$$

where $\tilde{\epsilon}_{T_g}(f_2)$ is estimated using a pseudo-label-based cross-entropy loss[8]:

$$
\tilde{\epsilon}_{T_g}(f_2) = -\mathbb{E}_{x_t \in T} \log f_2(l_h(g(x_t))|g(x_t))
$$

**Then, we define the discrepancy term based on CMD:**

$$
\begin{aligned}
\boldsymbol{L_{dis}} &= C_{S_g, T_g}(f_1, f_2, h) \\
&= \mathbb{E}_{x_t \sim T}[\mathrm{cmd}(f_1, f_2; g(x_t)) + \mathrm{cmd}(h, f_1; g(x_t))] + \mathbb{E}_{x_s \sim S}[\mathrm{cmd}(f_1, f_2; g(x_s)) - \mathrm{cmd}(h, f_2; g(x_s))]
\end{aligned}
$$

**Finally,** by introducing a trade-off parameter $\beta$ to balance the classification loss and the discrepancy, **the overall objective function can be written as follows:**

$$
\begin{cases}
\min_{f_1, f_2 \in H^F} L_{H_1} + L_{H_2} - \beta L_{dis} \\
\min_{h \in H^F, g} L_{H_1} + L_{H_2} + L_{ce} + \beta L_{dis}
\end{cases}
\tag{7}
$$

---

[8]In complex datasets, regularization terms are introduced to prevent $f_2$ from overfitting noisy pseudo-labels (A.1.3, Eq.8)

---

**Algorithm 1** MJE

---

**Input**: source data $S$ and unlabeled target data $T$
**Parameter**: constrained classifiers $f_1, f_2$; shared feature extractor $g$; upper bound classifier $h$; hyperparameters $\beta$; learning rate $\alpha$
**Output**: updated parameters $g, h$
**for** $iteration = 1, 2, \ldots$ **do**
$\quad$ **Step A**: optimize $g, f_1, f_2$ to satisfy the hypothesis constraint
$\quad\quad$ Compute $L_{H_1}, L_{H_2}$
$\quad\quad$ Update $g, f_1, f_2$:
$\quad\quad\quad$ $(g, f_1, f_2) \leftarrow (g, f_1, f_2) + \alpha\Delta(g, f_1, f_2)$ where $\Delta(g, f_1, f_2) = -\frac{\partial(L_{H_1}+L_{H_2})}{\partial(g, f_1, f_2)}$
$\quad$ **Step B**: maximize the discrepancy w.r.t. $f_1, f_2$ within the hypothesis space constraint
$\quad\quad$ Compute $L_{H_1}, L_{H_2}, L_{dis}$
$\quad\quad$ Update $f_1, f_2$:
$\quad\quad\quad$ $(f_1, f_2) \leftarrow (f_1, f_2) + \alpha\Delta(f_1, f_2)$ where $\Delta(f_1, f_2) = -\frac{\partial(L_{H_1}+L_{H_2}-\beta L_{dis})}{\partial(f_1, f_2)}$
$\quad$ **Step C**: minimize the entire target error upper bound w.r.t. $g, h$ for fixed $f_1, f_2$
$\quad\quad$ Compute $L_{ce}, L_{dis}$
$\quad\quad$ Update $g, h$:
$\quad\quad\quad$ $(g, h) \leftarrow (g, h) + \alpha\Delta(g, h)$, where $\Delta(g, h) = -\frac{\partial(L_{ce}+\beta L_{dis})}{\partial(g, h)}$
**end for**

---

The procedures described in Alg.1, during one training cycle, we first train $g, f_1, f_2$ to satisfy the hypothesis space constraint; second, we train $f_1, f_2$ to maximize the discrepancy term within the hypothesis space constraint; finally, we train $g, h$ to minimize the entire upper bound. Step C is executed four times in a training cycle, according to Saito et al. (2017b). See A.9 for further details on this objective.

## 4 Evaluation

In this section, we evaluate our proposed method using several different datasets (Digit (Netzer et al., 2011; LeCun et al., 1998; Hull, 1994), VisDA (Peng et al., 2017), Office-Home (Venkateswara et al., 2017), and Office-31 (Saenko et al., 2010)). We choose MCD (Saito et al., 2017b) as the baseline for a major comparison with ours because the two methods share the same concept of tightening the upper bound established by Ben-David et al. (2010). In addition, the two methods utilize the same network architecture and training procedure, enabling their results to be comparable. We conduct an ablation study (A.11) to demonstrate the contributions of each part of our proposal. We manually create an imbalanced label situation and demonstrate the robustness of our proposed method (A.10). The details of the experimental settings, hyperparameter selection, and training objectives are provided in A.1, A.7, and A.9, respectively.

### 4.1 Experiment on Digit Dataset

In this experiment, our proposed method was assessed in four types of adaptation scenarios by adopting commonly used digit datasets: MNIST (LeCun et al., 1998), street view house numbers (SVHN) (Netzer et al., 2011), and USPS (Hull, 1994), such that the results can be easily compared with other popular methods. All the experiments are performed in an unsupervised manner without any data augmentation. Tab. 1 lists the accuracies of the different methods. Our proposed method improves the performance in nearly all settings, except for a single result, compared with GPDA (Kim et al., 2019). However, their solution requires sampling, which increases the data size and is equivalent to adding Gaussian noise to the last layer of the classifier, which is considered a type of augmentation. The first setting of our proposal is $THS + L_1$, where $\gamma = \eta = 1$ serves as a direct comparison with MCD because the two methods share the same architecture, except for the upper bound, which demonstrates the importance of the joint error. The second and third settings of our proposal show the influence of the CMD and hypothesis space constraint, respectively.

---

[9]We use $\gamma = 0.1$ for MNIST $\rightarrow$ SVHN

Table 1: Accuracy of models adapted on digits datasets (* denotes the entire training set)

| METHOD | SVHN to MNIST | MNIST to SVHN | MNIST to USPS | MNIST* to USPS* | USPS to MNIST |
|---|---|---|---|---|---|
| Source Only | 67.1 | 21.3 | 76.7 | 79.7 | 63.4 |
| MDD[†](Long et al., 2015) | 71.1 | - | - | 81.1 | - |
| DANN[†]Ganin et al. (2016) | 71.1 | 25.1 | 77.3 | 85.1 | 73.2 |
| DRCN(Ghifary et al., 2016) | 82.0±0.1 | 40.1±0.1 | 91.8±0.1 | - | 73.7±0.1 |
| ADDA(Tzeng et al., 2017) | 76.0±1.8 | - | 89.4±0.2 | - | 90.1±0.8 |
| MCD(Saito et al., 2017b) | 96.2±0.4 | 11.2±1.1 | 94.2±0.7 | 96.5±0.3 | 94.1±0.3 |
| GPDA(Kim et al., 2019) | 98.2±0.1 | - | 96.5±0.2 | 98.1±0.1 | 96.4±0.1 |
| ours ($THS + L_1, \gamma = \eta = 1$) | 96.8±0.2 | 30.4±1.5[9] | 94.5±0.3 | 96.8±0.3 | 95.2±0.2 |
| ours ($THS + CMD, \gamma = \eta = 1$) | 97.5±0.2 | 31.5±1.8[9] | 95.3±0.3 | 97.2±0.2 | 95.6±0.2 |
| ours ($THS + CMD, \gamma = \eta = 0.9$) | 98.6±0.1 | 50.3±1.3 | 96.8±0.2 | 97.9±0.1 | 96.9±0.1 |

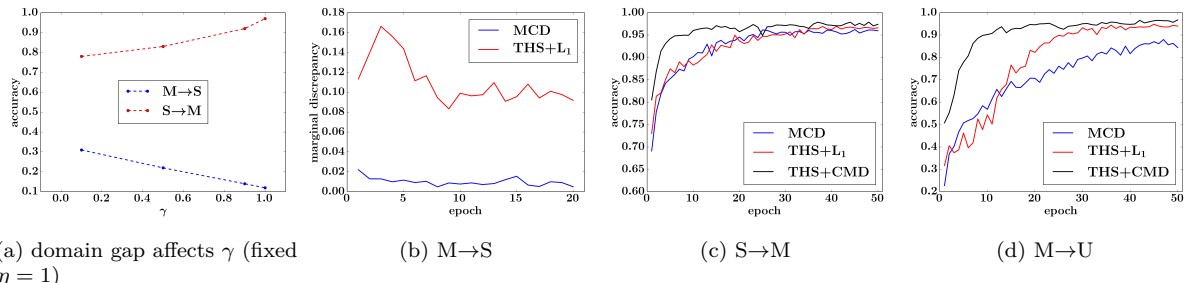

(a) domain gap affects $\gamma$ (fixed $\eta = 1$)

(b) M→S

(c) S→M

(d) M→U

Figure 4: (a) Sensitivity w.r.t. $\gamma$; (b) Comparisons for marginal discrepancy; (c)-(d) Comparisons for convergence rate; All plots are drawn based on the average of three runs.

Fig. 4a shows the result when we fix $\eta = 1$ and we can observe that the hyperparameter $\gamma$ is slightly sensitive to the domain gap. In summary, setting $\gamma = 1$ yields the best performance in most situations because $f_S, f_T$ can be considerably close after aligning the distributions, particularly in these easily adapted scenarios. However, in MNIST → SVHN, setting $\gamma = 0.1$ provides the optimum, which means that $f_S, f_T$ are far away owing to a large domain gap in which no feature extractor is capable of introducing an identical conditional distribution in the feature space. In addition, we test the performance on different values for $\gamma = \eta \neq 1$ setting and find that the accuracy remains high and varies subtly; therefore, we omit this figure. Furthermore, Fig. 4b empirically proves that simply minimizing the discrepancy between the marginal distributions does not necessarily lead to a reliable adaptation. In addition, Figs. 4c and 4d show the effectiveness of the CMD, which accelerates convergence and provides a slightly better result.

## 4.2 Experiment on VisDA Dataset

The VisDA dataset (Peng et al., 2017) is designed for a 12-class adaptation task from synthetic to real object images. The source domain contains 152,397 synthetic images generated by rendering 3D CAD models, whereas the target domain is collected from MSCOCO (Lin et al., 2014) and consists of 55,388 real images. Tab. 2 lists the accuracies of the different methods and the proposed method performed well under all settings. Images in this dataset are more complex than digits; however, our method provides reliable performance. Another key observation is that some competing methods (e.g., DANN and MCD), which can be categorized as distribution matching based on adversarial learning, perform worse than MDD (which simply matches statistics) in classes such as plane and horse, whereas our method performs better across all classes. This clearly demonstrates the importance of joint error.

Table 2: Accuracy of ResNet-101 model fine-tuned on VisDA dataset within 10 epochs

| METHOD | plane | bcycl | bus | car | horse | knife | mcycl | person | plant | sktbrd | train | truck | avg |
|---|---|---|---|---|---|---|---|---|---|---|---|---|---|
| Source Only | 55.1 | 53.3 | 61.9 | 59.1 | 80.6 | 17.9 | 79.7 | 31.2 | 81.0 | 26.5 | 73.5 | 8.5 | 52.4 |
| MDD(Long et al., 2015) | 87.1 | 63.0 | 76.5 | 42.0 | 90.3 | 42.9 | 85.9 | 53.1 | 49.7 | 36.3 | 85.8 | 20.7 | 61.1 |
| DANN(Ganin et al., 2016) | 81.9 | 77.7 | 82.8 | 44.3 | 81.2 | 29.5 | 65.1 | 28.6 | 51.9 | 54.6 | 82.8 | 7.8 | 57.4 |
| MCD(Saito et al., 2017b) | 87.0 | 60.9 | 83.7 | 64.0 | 88.9 | 79.6 | 84.7 | 76.9 | 88.6 | 40.3 | 83.0 | 25.8 | 71.9 |
| GPDA(Kim et al., 2019) | 83.0 | 74.3 | 80.4 | 66.0 | 87.6 | 75.3 | 83.8 | 73.1 | 90.1 | 57.3 | 80.2 | 37.9 | 73.3 |
| MCC(Jin et al., 2020) | 88.1 | 80.3 | 80.5 | 71.5 | 90.1 | 93.2 | 85.0 | 71.6 | 89.4 | 73.8 | 85.0 | 36.9 | 78.8 |
| ours ($THS + L_1, \gamma = \eta = 1$) | 86.3 | 82.7 | 83.7 | 68.7 | 87.9 | 72.7 | 85.4 | 61.5 | 87.3 | 55.5 | 75.2 | 34.1 | 73.4 |
| ours ($THS + CMD, \gamma = \eta = 1$) | 88.4 | 83.3 | 74.8 | 78.0 | 88.1 | 43.2 | 88.2 | 68.9 | 87.6 | 65.5 | 92.6 | 58.5 | 76.4 |
| ours ($THS + CMD, \gamma = \eta = 0.9$) | 93.8 | 79.5 | 79.3 | 55.9 | 93.9 | 93.8 | 86.5 | 80.3 | 91.6 | 87.7 | 85.4 | 51.6 | 81.6 |

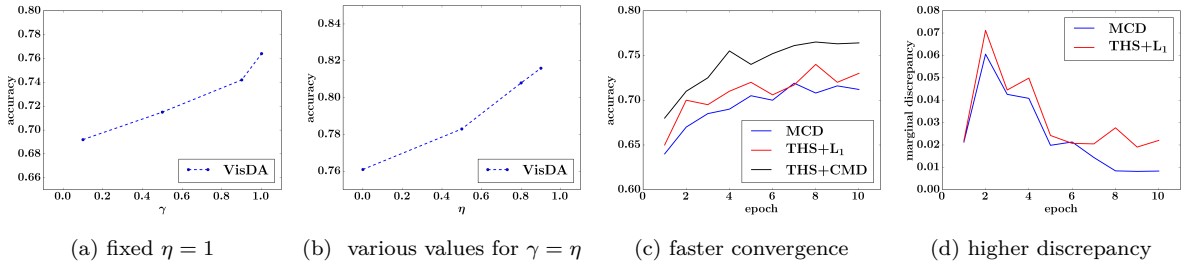

(a) fixed $\eta = 1$  (b) various values for $\gamma = \eta$  (c) faster convergence  (d) higher discrepancy

Figure 5: (a)-(b) Sensitivity w.r.t. $\gamma, \eta$; (c) Comparisons for convergence rate; (d) Comparisons for marginal discrepancy; All plots are drawn based on the average of three runs.

Fig. 5a shows the result when we fix $\eta = 1$ and the performance decreases while relaxing the constraint, which is perplexing to us. We expect an improvement here because it is difficult to believe that $f_S, f_T$ eventually exist in a similar space, determining from the relatively low prediction accuracy. Fig. 5b shows the adaptation performance of different values for $\gamma = \eta \neq 1$ setting, where the prediction accuracy decreases when $\eta$ is lower than 0.9. One possible reason is that $f_2$ and $h$ might nearly agree on the target domain, such that the prediction of $h$ cannot provide more accurate information on the target domain without introducing noisy pseudo-labels. Figs. 5c and 5d demonstrate the effectiveness of the cross-margin discrepancy and the importance of the joint error, respectively.

### 4.3 Experiment on Office-Home Dataset

Office-Home (Venkateswara et al., 2017) is a complex dataset containing 15,500 images from four significantly different domains: Art (Ar), Clipart (Cl), Product (Pr), and Real-World (Rw). Tab. 3 lists the accuracies of the different methods. The adaptation accuracy of the source only method is low, indicating that a large domain gap is likely to exist. In such cases, we believe that simply minimizing the discrepancy between the source and target domains can fail because the joint error may increase when aligning the distributions. Our proposal generally provides better performance when adaptation is difficult (1st, 8th, and 11th columns). These results suggest that our proposal, which incorporates the joint error into the target error upper bound, can boost performance, particularly when there is a large domain gap. SPL (Wang & Breckon, 2020) and SRDC (Tang et al., 2020) share a similar strategy, which can be described as cluster-based iterative pseudo-labeling. Despite their outstanding performances, these methods lack a theoretical guarantee of the generalization error bound. In addition, they assume that samples in the target domain are well clustered within the feature space and could be labeled by the Euclidean distance from clusters obtained using K-means. The first assumption is not always true because high-dimensional features can lie in a low-dimensional manifold (e.g., the Swiss Roll dataset), where no cluster exists. Regarding the second assumption, the labeling scheme can be vulnerable because K-means is sensitive to initialization, and the Euclidean distance can suffer from the curse of dimensionality such that there is little difference in the distances between different pairs of points.

Table 3: Accuracy of ResNet-50 model fine-tuned on the Office-Home dataset. We repeated each experiment five times and reported the average of the accuracy.

| METHOD | Ar→Cl | Ar→Pr | Ar→Rw | Cl→Ar | Cl→Pr | Cl→Rw | Pr→Ar | Pr→Cl | Pr→Rw | Rw→Ar | Rw→Cl | Rw→Pr | Avg |
|---|---|---|---|---|---|---|---|---|---|---|---|---|---|
| Source Only | 34.9 | 50.0 | 58.0 | 37.4 | 41.9 | 46.2 | 38.5 | 31.2 | 60.4 | 53.9 | 41.2 | 59.9 | 46.1 |
| DANN(Ganin et al., 2016) | 45.6 | 59.3 | 70.1 | 47.0 | 58.5 | 60.9 | 46.1 | 43.7 | 68.5 | 63.2 | 51.8 | 76.8 | 57.6 |
| MCD(Saito et al., 2017b) | 51.9 | 70.7 | 74.8 | 59.0 | 68.4 | 68.8 | 58.2 | 51.6 | 75.1 | 69.5 | 55.8 | 79.3 | 65.3 |
| CDAN(Long et al., 2018) | 50.7 | 70.6 | 76.0 | 57.6 | 70.0 | 70.0 | 57.4 | 50.9 | 77.3 | 70.9 | 56.7 | 81.6 | 65.8 |
| ADA(Wu et al., 2019) | 50.1 | 63.4 | 70.9 | 56.6 | 66.5 | 65.9 | 54.7 | 51.5 | 74.2 | 66.8 | 54.9 | 77.6 | 62.8 |
| SymNets(Zhang et al., 2019a) | 47.7 | 72.9 | 78.5 | 64.2 | 71.3 | 74.2 | 64.2 | 48.8 | 79.5 | 74.5 | 52.6 | 82.7 | 67.6 |
| SPL(Wang & Breckon, 2020) | 54.5 | 77.8 | 81.9 | 65.1 | 78.0 | 81.1 | 66.0 | 53.1 | 82.8 | 69.9 | 55.3 | 86.0 | 71.0 |
| AADA(Yang et al., 2020) | 54.0 | 71.3 | 77.5 | 60.8 | 70.8 | 71.2 | 59.1 | 51.8 | 76.9 | 71.0 | 57.4 | 81.8 | 67.0 |
| SRDC(Tang et al., 2020) | 52.3 | 76.3 | 81.0 | 69.5 | 76.2 | 78.0 | 68.7 | 53.8 | 81.7 | 76.3 | 57.1 | 85.0 | 71.3 |
| SCAL(Wang et al., 2022) | 55.3 | 72.7 | 78.7 | 63.1 | 71.7 | 73.5 | 61.4 | 51.6 | 79.9 | 72.5 | 57.8 | 81.0 | 68.3 |
| ours ($THS + CMD, \gamma = \eta = 0.9$) | 60.3 | 77.8 | 81.0 | 66.0 | 74.4 | 74.5 | 66.7 | 59.3 | 81.8 | 74.2 | 62.7 | 84.9 | 72.0 |

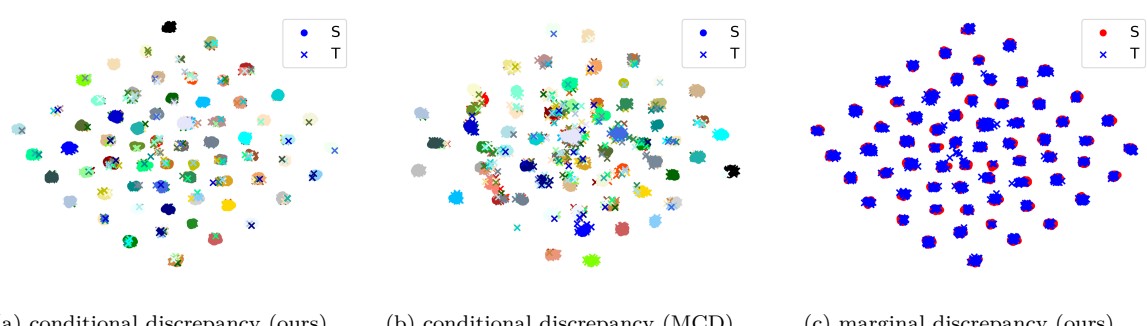

(a) conditional discrepancy (ours)  (b) conditional discrepancy (MCD)  (c) marginal discrepancy (ours)

Figure 6: Feature visualization with t-SNE in scenario Real to Art; (a)-(b) shows the conditional discrepancy by using different colors for each class; (c) shows the marginal discrepancy by using the same color for all the classes.

In addition, we plot the learned features using t-SNE (van der Maaten & Hinton, 2008) in Fig. 6, which shows Real to Art scenario. In Fig. 6a, we illustrate the source (dot) and target (cross) features using different colors that represent their classes. In the proposed method, most of the target features are clustered to their corresponding sources and do not show a large variance from the class center compared with MCD (Fig. 6b). We also visualize the features of the source (red dots) and target (blue crosses) domains. Fig. 6c shows the marginal discrepancy.

## 4.4 Experiment on Office-31 Dataset

Office-31 (Saenko et al., 2010) is a widely used dataset for verifying the effectiveness of a DA algorithm that contains three diverse domains, Amazon (A), Webcam (W), and DSLR (D), with 4,652 images in 31 unbalanced classes. Tab. 4 lists the results for Office-31. In tasks D→A and W→A, determining by the relatively low adaptation accuracy across all methods, a large domain gap between the source and target domains is likely to exist. Our method functions well in such cases, demonstrating that it penalizes the mismatch between the source and target domains. However, this advantage is not remarkable, particularly for the task A→W, where our proposed method shows relatively high variance and poor performance. One possible reason for this is that our method depends on learning reliable classifiers for the source domain to satisfy the constraints of the hypothesis space. However, the Amazon domain contains considerable noise, and the entire dataset lacks diversity (e.g., some domains only have several hundred samples with many duplicates), rendering it difficult for an error-bound-based method that generally requires a sufficient sample size to learn a reliable classifier.

Table 4: Accuracy of ResNet-50 model fine-tuned on the Office-31 dataset. We repeated each experiment five times and recorded the average and the standard deviation of the accuracy.

| METHOD | A→W | D→W | W→D | A→D | D→A | W→A | Avg |
|---|---|---|---|---|---|---|---|
| Source Only | 68.4±0.2 | 96.7±0.1 | 99.3±0.1 | 68.9±0.2 | 62.5±0.3 | 60.7±0.3 | 76.1 |
| DANN(Ganin et al., 2016) | 82.0±0.4 | 96.9±0.2 | 99.1±0.1 | 79.7±0.4 | 68.2±0.4 | 67.4±0.5 | 82.2 |
| ADDA(Tzeng et al., 2017) | 86.2±0.5 | 96.2±0.3 | 98.4±0.3 | 77.8±0.3 | 69.5±0.4 | 68.9±0.5 | 82.9 |
| MCD(Saito et al., 2017b) | 88.6±0.2 | 98.5±0.1 | 100.0±.0 | 92.2±0.2 | 69.5±0.1 | 69.7±0.3 | 86.5 |
| CDAN(Long et al., 2018) | 94.1±0.1 | 98.6±0.1 | 100.0±.0 | 92.9±0.2 | 71.0±0.3 | 69.3±0.3 | 87.7 |
| SymNets(Zhang et al., 2019a) | 90.8±0.1 | 98.8±0.3 | 100.0±0.0 | 93.9±0.5 | 74.6±0.6 | 72.5±0.5 | 88.4 |
| SPL(Wang & Breckon, 2020) | 92.7±0.0 | 98.1±0.0 | 99.8±0.0 | 93.7±0.0 | 76.4±0.0 | 76.9±0.0 | 89.6 |
| MCC(Jin et al., 2020) | 95.5±0.2 | 98.6±0.1 | 100.0±0.0 | 94.4±0.3 | 72.9±0.2 | 74.9±0.3 | 89.4 |
| SRDC(Tang et al., 2020) | 94.6±1.0 | 99.2±0.5 | 100.0±0.0 | 92.6±0.6 | 78.1±1.3 | 76.3±0.2 | 90.1 |
| SCAL(Wang et al., 2022) | 93.5±0.2 | 98.5±0.1 | 100.0±0.0 | 93.4±0.3 | 72.4±0.1 | 74.0±0.3 | 88.6 |
| ours ($THS + CMD, \gamma = \eta = 0.9$) | 91.9±0.5 | 99.0±0.2 | 100.0±.0 | 93.7±0.5 | 76.1±0.2 | 77.8±0.2 | 89.8 |

## 5 Conclusion

In this research, we propose a novel upper bound that considers joint errors. Subsequently, we pursue a tighter bound with reasonable constraints on the hypothesis space. Additionally, we adopt a novel cross-domain discrepancy for dissimilarity measurement, which alleviates instability during adversarial learning. Extensive empirical evidence shows that an invariant representation is not sufficient to guarantee good generalization performance in the target domain because the joint error affects, particularly when the domain gap is large. We believe that our results contribute significantly in understanding UDA and stimulate future work on the design of stronger adaptation algorithms.

## 6 Acknowledgements

This research is partially supported by JST Moonshot R&D Grant Number JPMJPS2011, CREST Grant Number JPMJCR2015 and Basic Research Grant (Super AI) of Institute for AI and Beyond of the University of Tokyo.

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

# A    Appendix

## A.1    Experimental Setting

### A.1.1    Digits Dataset

**SVHN $\leftrightarrow$ MNIST** We first examine the adaptation from SVHN (Fig. 7a) to MNIST (Fig. 7b). We used standard training and testing sets for both the source and target domains. The feature extractor contains three $5 \times 5$ convolutional layers with stride two $3 \times 3$ max pooling placed after the first two convolutional layers and a single fully-connected layer. For the classifiers, we used 2-layer fully-connected networks.

**MNIST $\leftrightarrow$ USPS** As for the adaptation between MNIST and USPS (Fig. 7c), we followed the training protocol established in Long et al. (2013) by sampling 2000 images from MNIST and 1800 from the USPS. For the test samples, the standard version was used for both source and target domains. The feature generator contains two $5 \times 5$ convolutional layers with stride two $2 \times 2$ max pooling placed after each convolutional layer and a single fully-connected layer. For the classifiers, we used 2-layer fully-connected networks.

Throughout the experiment, we employed the CNN architecture used in Saito et al. (2017b), where batch normalization was applied to each layer, and a 0.5 rate of dropout was used between fully-connected layers. In addition, spectral normalization (Miyato et al., 2018) was deployed for the classifiers in all subsequent experiments to stabilize adversarial learning. The major reason for utilizing this technique is that we observe a performance drop when the network is overtrained (this generally occurs after several hundreds of epochs). However, we are unsure whether this is caused by an overfitting of the source error or a general neural network training problem related to early stopping (this phenomenon can also be observed in other

algorithms). One common solution is to use a gradient reversal layer(Ganin et al., 2016) to balance the weight between the source error and discrepancy between the two domains. However, this introduces additional hyperparameters for tuning. Additionally, in the proposed CMD method, as illustrated in Fig. 8, we maximize $\log f_1(x, y_1) + \log(1 - f_2(x, y_1))$ with respect to the classifiers but minimize only $\log(1 - f_2(x, y_1))$ (a part of the objective) with respect to the feature extractor to avoid unnecessary oscillation during adversarial learning. This implies that the objectives we attempt to maximize and minimize are not is not equivalent. In such cases, it is difficult to employ a gradient reversal layer, which is intended to train the classifiers and feature extractors simultaneously in a single objective. With the help of spectral normalization, we can ensure that the classifiers are approximately Lipschitz such that a performance drop at an early stage can be avoided because the gradient with respect to the classifiers will be relatively small. Adam (Kingma & Ba, 2014) was used for optimization, with a minibatch size of 128 and a learning rate of $10^{-4}$.

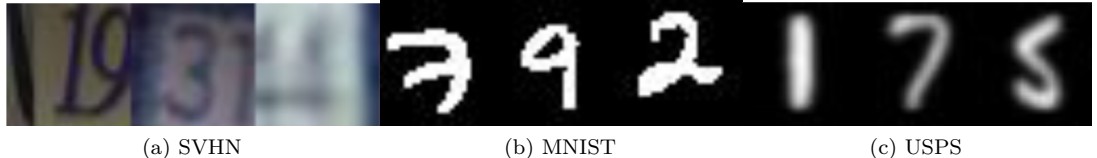

| (a) SVHN | (b) MNIST | (c) USPS |

Figure 7: Random samples from each dataset.

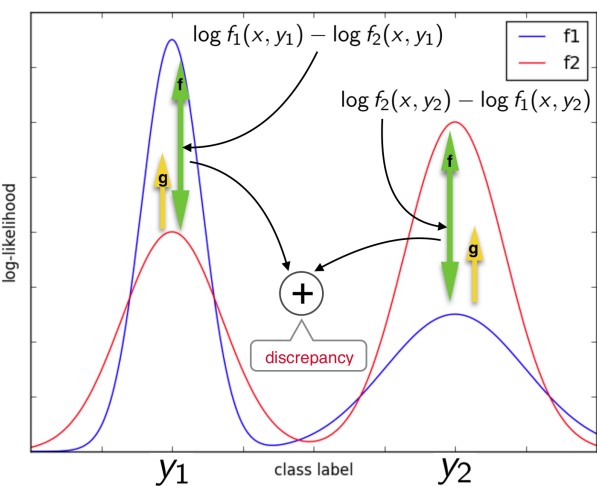

Figure 8: Illustrated details for the cross-margin discrepancy

### A.1.2 VisDA Dataset

The proposed method was further evaluated for object classification. The VisDA dataset (Peng et al., 2017), which is designed for a 12-class adaptation task from a synthetic object to real object images, was used. . The source domain contained 152,397 synthetic images generated by rendering 3D CAD models. Data from the target domain were collected from MSCOCO (Lin et al., 2014), which consisted of 55,388 real images. Because 3D models are generated without background or color diversity, the synthetic domain is slightly different from the real domain, rendering it a considerably more difficult problem than digit adaptation. In addition, this experiment was performed in an unsupervised fashion, and no data augmentation technique, excluding horizontal flipping, was allowed. By following the protocol established by Saito et al. (2017b), We evaluated our method by fine-tuning a ResNet-101 (He et al., 2015) model pretrained on ImageNet (Deng et al., 2009). The model, except for the last layer combined with a single-layer bottleneck, was used as a feature extractor, and a randomly initialized 2-layer fully-connected network was used as a classifier, where batch normalization was applied to each layer and a 0.5 dropout rate was conducted. Stochastic gradient

descent (SDG) with Nesterov moment was used for optimization with a minibatch size of 32 and an initial learning rate of $10^{-3}$, which decayed exponentially. The network architecture used in Saito et al. (2017b) was originally a 3-layer fully-connected classifier. However, as mentioned in the main paper, we have demonstrated that a relatively weaker classifier improves the stability of adversarial learning. Therefore, we leverage the framework proposed in Zhang et al. (2019b) by considering the top layer of the classifier, that is, a bottleneck, as part of the feature extractor (it does not change the entire complexity of the model). Another crucial point is that the network parameters with respect to ResNet-101 and the bottleneck were updated separately at a ratio of 1:10, which is the same as in Zhang et al. (2019b). A similar approach was used in Saito et al. (2017b) to decelerate the update of ResNet-101. The common reason is that excessively updating the feature extractor is not recommended because ResNet-101 is extremely powerful that it can easily overfit the source error. We applied horizontal flipping of the input images during training as the only data augmentation. For the hyperparameter, we tested $\gamma = \{0.1, 0.5, 0.9, 1\}$ and $\eta = \{0, 0.5, 0.8, 0.9\}$. For direct comparison, we reported the accuracy after 10 epochs.

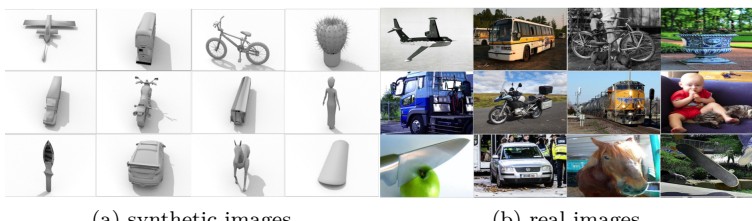

(a) synthetic images          (b) real images

Figure 9: (a) Samples from source domain. (b) Samples from target domain.

### A.1.3 Office-Home Dataset

Office-Home (Venkateswara et al., 2017) was a complex dataset (Fig. 10) containing 15,500 images from four significantly different domains: Art (paintings, sketches, and/or artistic depictions), Clipart (clip art images), Product (images without background), and Real-world (regular images captured with a camera). In this experiment, following the protocol of Zhang et al. (2019b), we evaluated our method by fine-tuning a ResNet-50 (He et al., 2015) model pretrained on ImageNet (Deng et al., 2009). The model, except for the last layer combined with a single-layer bottleneck, was used as a feature extractor, and a randomly initialized 2-layer fully-connected network with a width of 1024 was used as a classifier, where batch normalization was applied to each layer and a 0.5 dropout rate was conducted. For optimization, we used SGD with a Nesterov momentum term fixed at 0.9, where the batch size was 32, and the learning rate was adjusted according to Ganin et al. (2016). We applied horizontal flipping and resized the cropping of the input images during training as data augmentation, similar to that in Zhang et al. (2019b).

We introduce a slightly different loss function for $\tilde{\epsilon}_{T_g}(f_2)$: For unlabeled target data $\{x_t^i \in T\}_{i=1}^m$, the network predictions given by $h, f_2$ are represented by $\{q_i \in Q\}_{i=1}^m$ and $\{p_i \in P\}_{i=1}^m$, where the confidence of class $k$ is defined by

$$q_{i,k} = h(g(x_t^i), y = k), \quad p_{i,k} = f_2(g(x_t^i), y = k)$$

In addition to the hypothesis space constraint, we add a class balance regularization term for $f_2$ to encourage entropy maximization of the label distribution in the target domain, which improves the performance, particularly when the class number is large(Saito et al., 2017b):

$$\min_p \frac{1}{m} \sum_{i=1}^m KL(p_i \| q_i) + \sum_{k=1}^K \frac{\sum_{i=1}^m p_{i,k}}{m} \log \frac{\sum_{i=1}^m p_{i,k}}{m}$$

For a fixed $Q$, by setting the approximate gradient to zero, we obtain the following closed-form solution to the aforementioned problem(Dizaji et al., 2017):

$$p_{i,k}^* = \frac{q_{i,k}/\sqrt{\sum_{i=1}^m q_{i,k}}}{\sum_{k=1}^K q_{i,k}/\sqrt{\sum_{i=1}^m q_{i,k}}}$$

Finally, we train $g, f_2$ to push $p_{i,k}$ toward $p^*_{i,k}$ by minimizing the following KL divergence:

$$\tilde{\epsilon}_{T_g}(f_2) = -\frac{1}{m} \sum_{i=1}^{m} \sum_{k=1}^{K} p^*_{i,k} \log f_2(g(x^i_t), y = k) \tag{8}$$

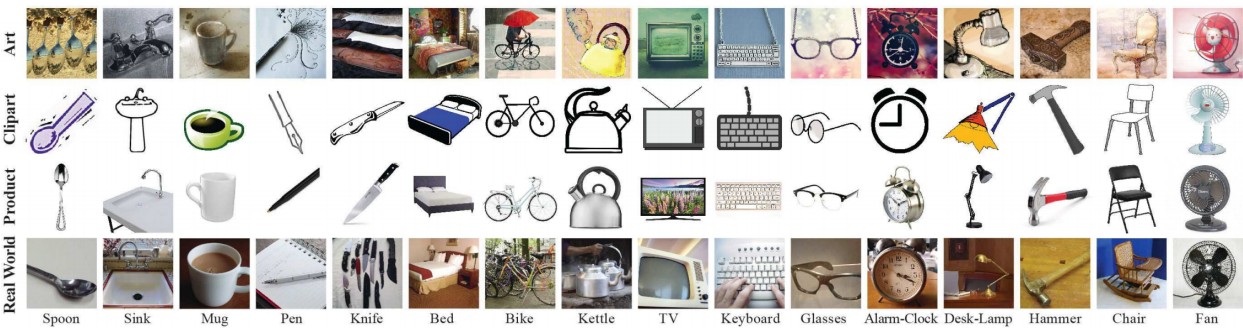

Figure 10: Sample images from the Office-Home dataset (Venkateswara et al., 2017).

### A.1.4 Office-31 Dataset

Office-31 (Saenko et al., 2010) (Fig. 11) is a widely used dataset for verifying the effectiveness of a DA algorithm. It contains three diverse domains: Amazon from the Amazon website, Webcam by web camera, and DSLR by digital SLR camera with 4,652 images in 31 unbalanced classes. Several noisy samples are shown in Fig.12 to backup our assumption for the failure case. In this experiment, following the protocol of Zhang et al. (2019b), we evaluated our method by fine-tuning a ResNet-50 (He et al., 2015) model pretrained on ImageNet (Deng et al., 2009). The model used here is nearly identical to that used in the Office-Home experiment, except for a different width of 2,048 for the classifiers. For optimization, we used SGD with a Nesterov momentum term fixed at 0.9, where the batch size was 32, and the learning rate was adjusted according to Ganin et al. (2016). We applied horizontal flipping and resized the cropping of the input images during training as data augmentation, similar to that in Zhang et al. (2019b).

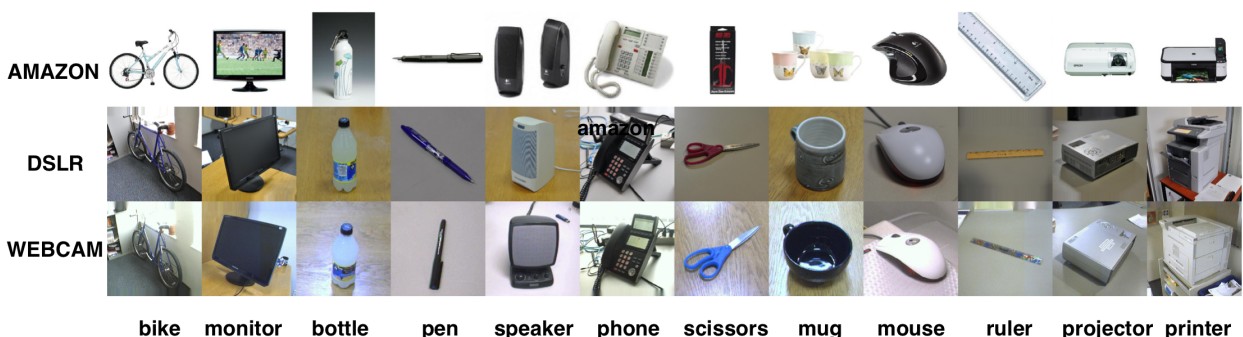

Figure 11: Sample images from the Office-31 dataset.

## A.2 Comparisons with Other Methods

This section present that, under certain conditions, the proposed joint error-based upper bound can be reduced to other popular upper bounds, demonstrating the generality of our proposal.

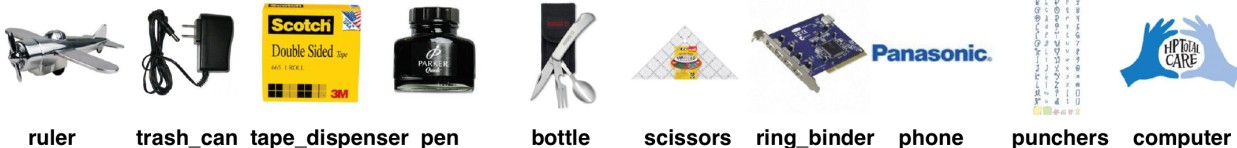

| ruler | trash_can | tape_dispenser | pen | bottle | scissors | ring_binder | phone | punchers | computer |

Figure 12: Noisy images from the Office-31 dataset.

### A.2.1 Margin Disparity Discrepancy

Zhang et al. (2019b) proposed a novel margin-aware generalization bound based on scoring functions and defined a new divergence MDD. The training objective used in MDD can be alternatively expressed as (here, $\epsilon(h, f)$ denotes the margin disparity) follows:

$$\min_{h \in H^F, g} [\epsilon_{S_g}(h) + \max_{f \in H^F} (\epsilon_{T_g}(f, h) - \epsilon_{S_g}(f, h))]$$

From Eq.3, if we set $f_1 = f_2 = f$ and free the constraint of $f$ to any $f \in H^F$, our proposal degrades exactly to MDD. As discussed previously, the assumption of an identical true labeling function $f_S = f_T$ is dangerous because we are not aware of the location of $f_T$ and a perfect alignment between the conditional distributions of the source and target domains is not likely to occur in practice. In addition, an unconstrained hypothesis space for $f$ is not helpful in building a tight bound.

### A.2.2 Maximum Classifier Discrepancy

Saito et al. (2017b) proposed two task-specific classifiers $f_1, f_2$ that are used to separate the decision boundary on the source domain, such that the feature extractor is encouraged to produce features near the support of the source samples. The objective used in MCD can be alternatively expressed as follows:

$$\begin{cases} \min_g [\epsilon_{S_g}(f_1) + \max_{f_1, f_2 \in H} (\epsilon_{T_g}(f_1, f_2))] \\ \text{s.t.} \quad H = \{f | \arg\min_{f \in H^F, g} \epsilon_{S_g}(f)\} \end{cases}$$

From Eq.3, if we set $\gamma = \eta = 1$ and $h = f_1$, our proposal reduces to MCD. As proved in 3.1, the proposed upper bound is optimized when $h = f_S$. However, this no longer holds after the upper bound is relaxed by obtaining the supremum; that is, setting $h = f_1$ does not necessarily minimize the objective. In addition, as discussed previously, assuming $H_2 = H_{sc} = H_1$ lacks generality because $f_T$ can be far from $f_S$ and does not necessarily classify all source samples, which means that the assumption of MCD is not likely to be applicable to cases in which a large domain gap exists.

### A.3 Rademacher Complexity

Let $H$ be a set of real-valued functions defined over set $X$. Given a sample $S \in X^m$, the empirical Rademacher complexity of $H$ is defined as follows:

$$\hat{\Re}_S(H) = \frac{2}{m} \mathbb{E}_\sigma \left[ \sup_{h \in H} | \sum_{i=1}^m \sigma_i h(x_i)| \Big| S = (x_1, ..., x_m) \right]$$

The expectation is calculated over $\sigma = (\sigma_1, ..., \sigma_n)$, where $\sigma_i$ is an independent uniform random variable that accepts values in $\{-1, +1\}$.

Rademacher complexity measures the ability of a class of functions to fit noise. It has the additional advantage of being data-dependent and can be measured from finite samples, which can lead to tighter bounds than those based on other measures of complexity, such as the VC-dimension. Following the established theory proposed by Mansour et al. (2009), we denote the empirical average of hypothesis $h : X \to \{0, 1\}$ by $\hat{R}(h)$ and its expectation over samples drawn according to the distribution considered by $R(h)$. The following is a version of the Rademacher complexity bound (Koltchinskii & Panchenko, 2000; Bartlett & Mendelson, 2002):

Let $H$ be a class of functions mapping $X \times Y \to [0,1]$ and $S = ((x_1, y_1), ..., (x_m, y_m))$ be a finite sample drawn i.i.d. according to the distribution $Q$. Then, for any $\sigma > 0$ with a probability of at least $1 - \sigma$ over a sample $S$ of size $m$, the following inequality holds for all $h \in H$:

$$R(h) \leq \hat{R}(h) + \hat{\Re}_S(H) + 3\sqrt{\frac{\log \frac{2}{\sigma}}{2m}}$$

From our proposed upper bound, we have: $\epsilon_T(h) \leq \epsilon_S(h) + C_{S,T}(f_S, f_T, h)$, where $C_{S,T}$ is further bounded by: $\sup_{f_1 \in H_1, f_2 \in H_2}\{\epsilon_T(f_1, f_2) + \epsilon_S(f_1, f_2) + \epsilon_T(h, f_1) - \epsilon_S(h, f_2)\}$. We relax this bound by applying $\epsilon_S(f_1, f_2) \leq \epsilon_S(f_1, h) + \epsilon_S(f_2, h)$ and name it $d(S, T; h) = \sup_{f_1 \in H_1, f_2 \in H_2}\{\epsilon_T(f_1, f_2) + \epsilon_S(f_1, h) + \epsilon_T(f_1, h)\}$. We assume that the loss function $\epsilon$ is bounded by $M > 0$:$\epsilon(h, h') \leq M$ for all $h, h' \in H$. Let $Q$ be a distribution over $X$ and let $\hat{Q}$ denote the corresponding empirical distribution for sample $S = (x_1, ..., x_m)$. Then, we can scale the loss $\epsilon$ to $[0,1]$ by dividing by $M$ and denote the new class by $\epsilon_{H_1, H_2}$, which represents the class of functions $\{x \to \epsilon(f_1(x), f_2(x)) : f_1 \in H_1, f_2 \in H_2\}$. From the aforementioned theorem, for any $\sigma > 0$ with a probability of at least $1 - \sigma$, the following inequality holds for all $f_1 \in H_1, f_2 \in H_2$:

$$\frac{\epsilon_Q(f_1, f_2)}{M} \leq \frac{\epsilon_{\hat{Q}}(f_1, f_2)}{M} + \hat{\Re}_S(\epsilon_{H_1, H_2}/M) + 3\sqrt{\frac{\log \frac{2}{\sigma}}{2m}}$$

The empirical Rademacher complexity has the property that $\hat{\Re}(\alpha H) = \alpha \hat{\Re}(H)$ for any hypothesis class $H$ and positive number $\alpha$ simplifies the aforementioned bound. Let $S$ be a distribution over $X$ and $\hat{S}$ be the corresponding empirical distribution for sample $\tilde{S}$, and let $T$ be a distribution over $X$ and $\hat{T}$ be the corresponding empirical distribution for sample $\tilde{T}$. Then, for any $\sigma > 0$, with a probability of at least $1 - \sigma$ over samples $\tilde{S}$ of size $m$ drawn according to $S$ and samples $\tilde{T}$ of size $n$ drawn according to $T$, we can write the following:

$$\begin{aligned}
d(S, T; h) \leq &\sup_{f_1 \in H_1, f_2 \in H_2}\{\epsilon_{\hat{T}}(f_1, f_2) + \epsilon_{\hat{S}}(f_1, h) + \epsilon_{\hat{T}}(f_1, h)\} \\
&+ \hat{\Re}_{\tilde{T}}(\epsilon_{H_1, H_2}) + \hat{\Re}_{\tilde{S}}(\epsilon_{H_1}) + \hat{\Re}_{\tilde{T}}(\epsilon_{H_1}) \\
&+ 3M(\sqrt{\frac{\log \frac{2}{\sigma}}{2m}} + 2\sqrt{\frac{\log \frac{2}{\sigma}}{2n}})
\end{aligned}$$

## A.4 Compatibility

Despite the assumption that $H$ includes $f_T$ was made in previous research like Mansour et al. (2009), we agree that assuming the true labeling functions lie in a specific hypothesis space lacks generality. However, if the algorithm is run within finite samples, it is possible that a function inside a specific hypothesis space with enough complexity can perfectly mimic the behavior of the true labeling function for those samples. Therefore, even if the hypothesis space we use does not contain true labeling functions, it does not harm the actual learning process. The proof is as follows:

$$\begin{aligned}
\epsilon_T(h) &= \epsilon_T(h, f_T) \\
&= \epsilon_T(h, f_T) - \epsilon_T(h, f_S) + \epsilon_T(h, f_S) + \epsilon_S(h, f_S) - \epsilon_S(h, f_S) + \epsilon_S(h, f_T) - \epsilon_S(h, f_T) \\
&= \epsilon_S(h, f_S) + (\epsilon_T(h, f_T) - \epsilon_T(h, f_S)) + (\epsilon_S(h, f_T) - \epsilon_S(h, f_S)) + \epsilon_T(h, f_S) - \epsilon_S(h, f_T) \\
&\leq \epsilon_S(h) + \epsilon_T(f_S, f_T) + \epsilon_S(f_S, f_T) + \epsilon_T(h, f_S) - \epsilon_S(h, f_T) \\
&\leq \epsilon_S(h) + \epsilon_T(f_S, f_S^*) + \epsilon_T(f_S^*, f_T^*) + \epsilon_T(f_T^*, f_T) + \epsilon_S(f_S, f_S^*) + \epsilon_S(f_S^*, f_T^*) + \epsilon_S(f_T^*, f_T) \\
&\quad + \epsilon_T(h, f_S^*) + \epsilon_T(f_S^*, f_S) - \epsilon_S(h, f_T^*) + \epsilon_S(f_T^*, f_T) \\
&= \epsilon_S(h) + C_{S,T}(f_S^*, f_T^*, h) + \theta,
\end{aligned}$$

where $\theta = 2\epsilon_T(f_S, f_S^*) + \epsilon_S(f_S, f_S^*) + 2\epsilon_S(f_T^*, f_T) + \epsilon_T(f_T^*, f_T)$.

Let $f_S^* = \arg\min_{f \in H} \epsilon_{S \bigcup 2T}(f_S, f)$ and $f_T^* = \arg\min_{f \in H} \epsilon_{2S \bigcup T}(f_T, f)$ such that $f_S^*, f_T^* \in H$ can perfectly mimic the behavior of $f_S, f_T$ on $S$ and $T$. Then $\theta$ can be completely ignored, and the obtained bound becomes compatible with the original upper bound by replacing $f_S, f_T$ with $f_S^*, f_T^*$.

### A.5 Validity

In Sec.3.2, we assume that it is possible to create two subspaces $H_1, H_2 \subseteq H_f$ such that:

$$C_{S,T}(f_S, f_T, h) \leq \max_{f_1 \in H_1, f_2 \in H_2} C_{S,T}(f_1, f_2, h)$$

This assumption is difficult to prove theoretically; thus, we show the validity of the inequality using experimental results. In this section, we select an adaptation scenario in which the domain gap is large (Product to Clipart scenario of the Office-Home dataset). We used the full source and target labels to estimate $C_{S,T}(f_S, f_T, h)$ as the ground truth. The upper bound is computed with two subspaces $H_1, H_2$ given by the target-driven hypothesis space proposal (Sec.3.2.1) that utilizes pseudo-labels, as follows:

$$\begin{cases} H_1 = \{f_1 | \arg\min_{f_1 \in H}[\epsilon_S(f_1)]\} \\ H_2 = \{f_2 | \arg\min_{f_2 \in H}[\gamma\epsilon_S(f_2) + (1 - \eta)\tilde{\epsilon}_T(f_2)]\} \end{cases}$$

Fig.13 shows that our proposal remains a valid upper bound in practice even if the domain gap is extremely large that the subspace $H_2$ is unlikely to include $f_T$.

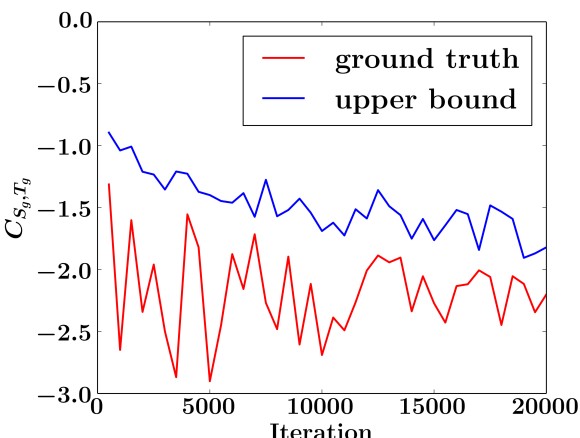

Figure 13: Estimated means of ground truth and upper bound by three runs in scenario Pr→Cl adapted with $THS + CMD$ proposal.

### A.6 Proof

#### A.6.1 Proof for the Upper Bound

Given the aforementioned notation and the triangle inequality, we have the following:

$$\begin{aligned} \epsilon_T(h) &= \epsilon_T(h, f_T) \\ &= \epsilon_T(h, f_T) + [\epsilon_T(h, f_S) - \epsilon_T(h, f_S)] + [\epsilon_S(h, f_S) - \epsilon_S(h, f_S)] + [\epsilon_S(h, f_T) - \epsilon_S(h, f_T)] \\ &= \epsilon_S(h, f_S) + [\epsilon_T(h, f_T) - \epsilon_T(h, f_S)] + [\epsilon_S(h, f_T) - \epsilon_S(h, f_S)] + \epsilon_T(h, f_S) - \epsilon_S(h, f_T) \\ &\leq \epsilon_S(h) + \epsilon_T(f_S, f_T) + \epsilon_S(f_S, f_T) + \epsilon_T(h, f_S) - \epsilon_S(h, f_T) \end{aligned}$$

#### A.6.2 Proof for the Optimum

We demonstrate that our upper bound is minimized when $h = f_S$ and is equivalent to $\epsilon_T(f_S, f_T)$.

First, by setting $h = f_S$, we obtain

$$\epsilon_S(h) + \epsilon_T(f_S, f_T) + \epsilon_S(f_S, f_T) + \epsilon_T(h, f_S) - \epsilon_S(h, f_T) = \epsilon_T(f_S, f_T)$$

Then, according to the triangle inequality and nonnegativity of the distance metric, we have

$$
\begin{aligned}
[\epsilon_S(h) + \epsilon_S(f_S, f_T)] + \epsilon_T(f_S, f_T) + \epsilon_T(h, f_S) - \epsilon_S(h, f_T) \\
\geq \epsilon_S(h, f_T) + \epsilon_T(f_S, f_T) + \epsilon_T(h, f_S) - \epsilon_S(h, f_T) \\
= \epsilon_T(f_S, f_T) + \epsilon_T(h, f_S) \\
\geq \epsilon_T(f_S, f_T)
\end{aligned}
$$

### A.6.3 Proof for the Relation to Joint error

We demonstrate that when our upper bound is minimized, it is equivalent to the upper bound of the optimal joint error $\lambda$ because of the following:

$$
\begin{aligned}
\epsilon_T(f_S, f_T) &= \epsilon_T(f_S, f_T) + \epsilon_S(f_S, f_S) \\
&= \epsilon_T(f_S) + \epsilon_S(f_S) \\
&\geq \min_{h \in H}(\epsilon_T(h) + \epsilon_S(h)) = \lambda
\end{aligned}
$$

### A.7 Hyperparameter Selection

In all experiments, the hyperparameters $\gamma, \eta$ were set based on the validation performance on a separate dataset consisting of a few hundred labeled target samples. As shown in the main paper, the performance of our proposed method is sensitive to changes in the hyperparameters. However, it also suggests that an appropriate choice of hyperparameters can significantly boost the performance, which enabled us to select the hyperparameters via a validation dataset with 200 samples. Fig. 14 shows two examples of the validation performance of different adaptation tasks with respect to $\gamma, \eta$.

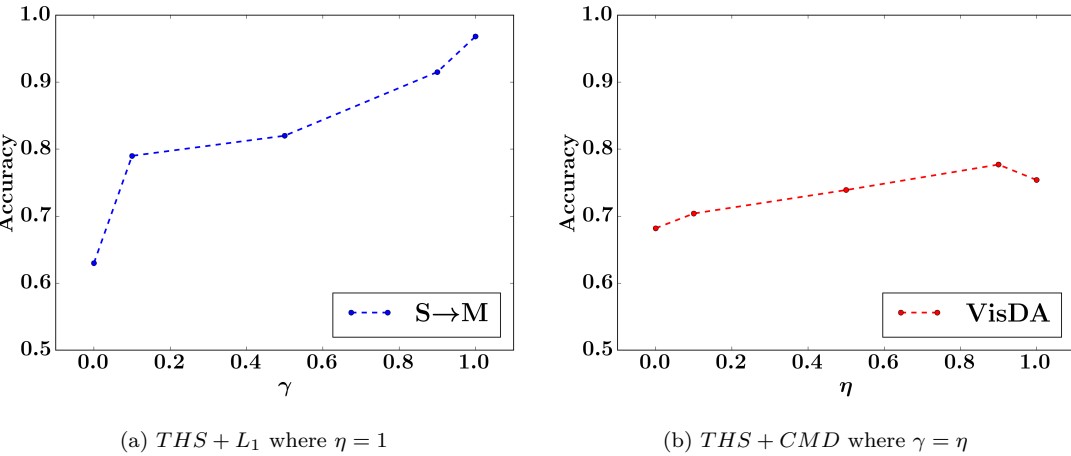

(a) $THS + L_1$ where $\eta = 1$        (b) $THS + CMD$ where $\gamma = \eta$

Figure 14: Validation performance when varying the hyperparameters $\gamma, \eta$ on a separate dataset

### A.8 Relations between the Margin Theory and CMD

The margin between the data points and the classification surface plays a significant role in achieving strong generalization performance. To quantify $\epsilon$ into a differentiable measurement as a surrogate for the 0-1 loss, we introduce the margin theory developed by Koltchinskii & Panchenko (2002), where the margin loss is interpreted as follows:

$$
\mathbb{E}_{(x,y) \in D}[\max(0, 1 + \max_{y' \neq y} s(y'|x) - s(y|x))],
$$

where the output of the score function $s(y|x)$ for multiclass classification indicates the confidence of the prediction for class $y$.

We aim to utilize this concept to further improve the reliability of our proposed method by leveraging this margin loss to define a novel measurement of the discrepancy between the two hypotheses $f_1, f_2$ (e.g., a multilayer perceptron with an output layer of a softmax function) over a distribution $D$, namely, the CMD:

$$\epsilon_D(f_1, f_2) = \mathbb{E}_{x \in D}[\text{cmd}(f_1, f_2; x)]$$

Before further discussion, we first introduce two distributions $D_{f_1}, D_{f_2}$ induced by labeling functions $f_1, f_2$, respectively, where $D_{f_1} = \{(x, l_{f_1}(x))|x \sim D\}$ and $D_{f_2} = \{(x, l_{f_2}(x))|x \sim D\}$ (function $l$ returns the predictive labels). Then, we consider the case in which the two hypotheses ($f_1$ and $f_2$) disagree, that is, $y_1 = l_{f_1}(x) \neq l_{f_2}(x) = y_2$, and the primitive loss is defined as follows:

$$\begin{aligned}
\text{cmd}(f_1, f_2; x) &= \log f_1(y_1|x) - \log f_2(y_1|x) + \log f_2(y_2|x) - \log f_1(y_2|x) \\
&= \log f_1(y_1|x) - \log f_1(y_2|x) + \log f_2(y_2|x) - \log f_2(y_1|x)
\end{aligned}$$

Then, the CMD $\epsilon_D(f_1, f_2)$ can be expressed as follows:

$$\mathbb{E}_{(x,y) \in D_{f_2}}[\max_{y' \neq y} \log f_1(y'|x) - \log f_1(y|x)] + \mathbb{E}_{(x,y) \in D_{f_1}}[\max_{y' \neq y} \log f_2(y'|x) - \log f_2(y|x)]$$

This is the sum of the margin loss for $f_1$ on $D_{f_2}$ and the margin loss for $f_2$ on $D_{f_1}$ if we use the logarithm of softmax as the score function.

Following the method introduced by Goodfellow et al. (2014) for mitigating the burden of exploding or vanishing gradients, we optimize the dual form of the aforementioned primitive loss in practice:

$$\text{cmd}(f_1, f_2; x) = \log f_1(y_1|x) + \log(1 - f_1(y_2|x)) + \log f_2(y_2|x) + \log(1 - f_2(y_1|x))$$

## A.9 Details of the Training Objective

This section briefly describes the training objectives of the proposed method. We consider the THS objective as an example, and each term in the objective is defined as follows:

$$\begin{cases}
\min_{h \in H^F, g}[\epsilon_{S_g}(h) + \max_{f_1 \in H_1, f_2 \in H_2} C_{S_g, T_g}(f_1, f_2, h)] \\
\text{s.t.} \quad H_1, H_2 = \{f_1, f_2 | \arg\min_{f_1, f_2 \in H^F, g}[\epsilon_{S_g}(f_1) + \gamma \epsilon_{S_g}(f_2) + (1 - \eta)\tilde{\epsilon}_{T_g}(f_2)]\},
\end{cases}$$

where $g$ represents a feature extractor and $h, f_1, f_2$ are classifiers with a softmax output (i.e., $h(x, y)$ represents the confidence of sample $x$ classified as label $y$). First, we can use the cross-entropy loss to achieve the following hypothesis space constraint:

$$\begin{cases}
\epsilon_{S_g}(f_1) = -\mathbb{E}_{x,y \sim S} \log(f_1(y|g(x))) \\
\gamma \epsilon_{S_g}(f_2) + (1 - \eta)\tilde{\epsilon}_{T_g}(f_2) = -\gamma \mathbb{E}_{x,y \sim S} \log(f_2(y|g(x))) - (1 - \eta)\mathbb{E}_{x \sim T} \log(f_2(l_h(x)|g(x)))
\end{cases}$$

We then use the proposed CMD to reorganize $C_{S_g, T_g}(f_1, f_2, h)$. Before further derivation, we introduce three proxy distributions $S_{f_1=f_2}, S_{f_1 \setminus f_2}, and S_{f_2 \setminus f_1}$ induced by $f_1, f_2$, where $S_{f_1=f_2} = \{x, y|x \sim S, l_{f_1}(x) = l_{f_2}(x) = y\}, S_{f_1 \setminus f_2} = \{x, y|x \sim S, y = l_{f_1}(x) \neq l_{f_2}(x)\}, S_{f_2 \setminus f_1} = \{x, y|x \sim S, l_{f_1}(x) \neq l_{f_2}(x) = y\}$ (the labeling function $l$ returns labels of the most confident prediction by $f$). Analogously, we define the others as

Table 5: Influence of $\beta$ on the adaptation of SVHN→MNIST under our proposal ($THS + CMD, \eta = 0$).

| SVHN to MNIST | $\beta = 0.001$ | $\beta = 0.01$ | $\beta = 0.1$ |
|---|---|---|---|
| Acc | 98.4±0.1 | 98.6±0.1 | 97.2±0.3 |

$S_{f_1=h}, S_{f_2=h}, T_{f1\setminus f2}$. According to the definition of CMD (Sec.3.3):

$$
\begin{aligned}
&\check{C}_{S_g,T_g}(f_1, f_2, h)\\
&= \mathbb{E}_{x,y\sim S_{f_1=f_2}}[\log\max(f_1(y|g(x)), f_2(y|g(x))) + \log\max(1 - f_1(y|g(x)), 1 - f_2(y|g(x)))]\\
&+ \mathbb{E}_{x,y\sim S_{f_1\setminus f_2}}[\log f_1(y|g(x)) + \log(1 - f_2(y|g(x)))] + \mathbb{E}_{x,y\sim S_{f_2\setminus f_1}}[\log(1 - f_1(y|g(x))) + \log f_2(y|g(x))]\\
&+ \mathbb{E}_{x,y\sim T_{f_1=f_2}}[\log\max(f_1(y|g(x)), f_2(y|g(x))) + \log\max(1 - f_1(y|g(x)), 1 - f_2(y|g(x)))]\\
&+ \mathbb{E}_{x,y\sim T_{f_1\setminus f_2}}[\log f_1(y|g(x)) + \log(1 - f_2(y|g(x)))] + \mathbb{E}_{x,y\sim T_{f_2\setminus f_1}}[\log(1 - f_1(y|g(x))) + \log f_2(y|g(x))]\\
&+ \mathbb{E}_{x,y\sim T_{f_1=h}}[\log\max(f_1(y|g(x)), h(y|g(x))) + \log\max(1 - f_1(y|g(x)), 1 - h(y|g(x)))]\\
&+ \mathbb{E}_{x,y\sim T_{f_1\setminus h}}[\log f_1(y|g(x)) + \log(1 - h(y|g(x)))] + \mathbb{E}_{x,y\sim T_{h\setminus f_1}}[\log(1 - f_1(y|g(x))) + \log h(y|g(x))]\\
&- \mathbb{E}_{x,y\sim S_{f_2=h}}[\log\max(h(y|g(x)), f_2(y|g(x))) + \log\max(1 - h(y|g(x)), 1 - f_2(y|g(x)))]\\
&- \mathbb{E}_{x,y\sim S_{f_2\setminus h}}[\log f_2(y|g(x)) + \log(1 - h(y|g(x)))] - \mathbb{E}_{x,y\sim S_{h\setminus f_2}}[\log(1 - f_2(y|g(x))) + \log h(y|g(x))],
\end{aligned}
\tag{9}
$$

where part of this objective can be regarded as a CGAN objective aiming to align the conditional distributions of three pairs of hypothesis induced distributions ($S_{f_1=f_2} \Leftrightarrow T_{f_1=f_2}$, $S_{f_1\setminus f_2} \Leftrightarrow T_{f_2\setminus f_1}$, $S_{f_2\setminus f_1} \Leftrightarrow T_{f_1\setminus f_2}$) under a minor assumption that $f_1, f_2$ are more confident about $S, T$, respectively.

As explained in A.1.1, to avoid unnecessary oscillations during adversarial learning, we only optimize a part of the objective (the part considered as fake by the classifier) w.r.t. $g, h$ such that it can be consistent with the general CGAN objective. This implies the objective ($\hat{C}_{S_g,T_g}(f_1, f_2, h)$) that we attempt to maximize is not equivalent to ($\check{C}_{S_g,T_g}(f_1, f_2, h)$) that we attempt to minimize.

$$
\begin{aligned}
&\check{C}_{S_g,T_g}(f_1, f_2, h)\\
&= \mathbb{E}_{x,y\sim S_{f_1=f_2}}\log\max(1 - f_1(y|g(x)), 1 - f_2(y|g(x)))\\
&+ \mathbb{E}_{x,y\sim S_{f_1\setminus f_2}}\log(1 - f_2(y|g(x))) + \mathbb{E}_{x,y\sim S_{f_2\setminus f_1}}\log(1 - f_1(y|g(x)))\\
&+ \mathbb{E}_{x,y\sim T_{f_1=f_2}}\log\max(1 - f_1(y|g(x)), 1 - f_2(y|g(x)))\\
&+ \mathbb{E}_{x,y\sim T_{f_1\setminus f_2}}\log(1 - f_2(y|g(x))) + \mathbb{E}_{x,y\sim T_{f_2\setminus f_1}}\log(1 - f_1(y|g(x)))\\
&+ \mathbb{E}_{x,y\sim T_{f_1=h}}\log\max(1 - f_1(y|g(x)), 1 - h(y|g(x)))\\
&+ \mathbb{E}_{x,y\sim T_{f_1\setminus h}}\log(1 - h(y|g(x))) + \mathbb{E}_{x,y\sim T_{h\setminus f_1}}\log(1 - f_1(y|g(x)))\\
&- \mathbb{E}_{x,y\sim S_{f_2=h}}\log\max(1 - h(y|g(x)), 1 - f_2(y|g(x)))\\
&- \mathbb{E}_{x,y\sim S_{f_2\setminus h}}\log(1 - h(y|g(x))) - \mathbb{E}_{x,y\sim S_{h\setminus f_2}}\log(1 - f_2(y|g(x))),
\end{aligned}
\tag{10}
$$

Because we apply different measurements to the source error and discrepancy, we introduce a scaling factor $\beta$ to ensure that neither of them can dominate back propagation. $\beta = 0.01$ was used in all the experiments because we observed that the source error was generally approximately $1e - 2$ and the discrepancy was generally approximately $1e0$. We also conducted a simple experiment to check the influence of $\beta$ and the results are listed in Tab.5.

## A.10 Imbalance Label Distribution

In this section, we describe an additional experiment in which the label distribution of the source is significantly different from that of the target domain. We choose Office-Home dataset and manually remove the samples of an entire class ('clock') from the target (Clipart) domain to create an imbalance scenario. The adaptation accuracy is listed in Tab.6. In theory, our proposal does not suffer from an imbalance in the label distribution. Unlike other methods that can only match marginal distributions, the proposed method is designed to align conditional distributions. In practice, the experimental results demonstrate the validity of our theory because no remarkable performance drop exists in our proposed method compared to other methods. Despite this

Table 6: Accuracy of ResNet-50 model fine-tuned on the Office-Home dataset. * represents the imbalance label distribution setting, where we manually remove the samples of an entire class from the target domain.

| METHOD | Ar→Cl | Pr→Cl | Rw→Cl |
|---|---|---|---|
| Source Only | 34.9 | 31.2 | 41.2 |
| DANN(Ganin et al., 2016) | 45.6 | 43.7 | 51.8 |
| MCD(Saito et al., 2017b) | 51.9 | 51.6 | 55.8 |
| CDAN(Long et al., 2018) | 50.7 | 50.9 | 56.7 |
| SymNets(Zhang et al., 2019a) | 47.7 | 48.8 | 52.6 |
| SPL(Wang & Breckon, 2020) | 54.5 | 53.1 | 55.3 |
| AADA(Yang et al., 2020) | 54.0 | 51.8 | 57.4 |
| SRDC(Tang et al., 2020) | 52.3 | 53.8 | 57.1 |
| SCAL(Wang et al., 2022) | 55.3 | 51.6 | 57.8 |
| ours $(THS + CMD, \eta = 0.9)$ | 60.3 | 59.2 | 62.7 |
| DANN* | 43.1 | 40.5 | 49.2 |
| MCD* | 50.5 | 49.7 | 54.3 |
| ours* $(THS + CMD, \eta = 0.9)$ | 60.0 | 58.5 | 62.2 |

Table 7: Simple ablation study to demonstrate the contribution from each part of the proposal ($\sqrt{}$ in the column "optimal joint error" means to include the joint error in the upper bound, in contrast to the training objective proposed by MCD; $\sqrt{}$ in the column "CMD" means to use the proposed discrepancy measurement for $\epsilon$, whereas $\times$ indicates that we use $L_1$ norm alternatively; $\sqrt{}$ in the column "pseudo-labels induced hypothesis space" means the hypothesis space for $f_2$ is created based on pseudo-labels of target samples, which is corresponding to our THS proposal in the main paper, where $\eta \neq 1$). We repeat the adaptation from SVHN to MNIST five times and record the average and the standard deviation of the accuracy.

| METHOD | OPTIMAL JOINT ERROR | CROSS MARGIN DISCREPANCY | PSEUDO-LABELS INDUCED HYPOTHESIS SPACE | SVHN TO MNIST |
|---|---|---|---|---|
| BASELINE | $\times$ | $\times$ | $\times$ | 96.2±0.4 |
|  | $\times$ | $\sqrt{}$ | $\times$ | 96.6±0.2 |
|  | $\times$ | $\times$ | $\sqrt{}$ | 97.1±0.3 |
|  | $\times$ | $\sqrt{}$ | $\sqrt{}$ | 97.6±0.2 |
| OURS | $\sqrt{}$ | $\times$ | $\times$ | 96.8±0.2 |
|  | $\sqrt{}$ | $\sqrt{}$ | $\times$ | 97.5±0.2 |
|  | $\sqrt{}$ | $\times$ | $\sqrt{}$ | 98.2±0.2 |
|  | $\sqrt{}$ | $\sqrt{}$ | $\sqrt{}$ | 98.6±0.1 |

extreme imbalance in the label distribution, our proposed method still outperformed other methods trained on the full target domain.

## A.11 Ablation Study

In this section, we conduct a simple ablation study to show the way each part of the proposal, that is, the objective, including the joint error, the CMD, and the pseudo-label-induced hypothesis space, contributes to the performance gain. For comparison, we chose MCD(Saito et al., 2017b) as the baseline because its assumption regarding the hypothesis space is similar to that of our proposal. In addition, the two methods shared the same network architecture in their implementation, rendering the experimental results directly comparable. In Tab. 7, the fifth row shows the effectiveness of the proposed upper bound, where we use $L_1$ as the discrepancy measurement without involving any pseudo-target information, and improves the performance compared to MCD in the first row. The sixth row shows the results when it is replaced with the CMD. The last row shows the results of leveraging information from the pseudo-labeled target samples to create a more reliable hypothesis space for $f_2$. These results show that every part of our proposal improves the performance. To verify the effectiveness of the proposed method further, we conducted another ablation

Table 8: Simple ablation study to demonstrate the contribution from each part of the proposal ($\sqrt{}$ in the column "optimal joint error" means to include the joint error in the upper bound, in contrast to the training objective proposed by MCD; $\sqrt{}$ in the column "CMD" means to use the proposed discrepancy measurement for $\epsilon$, whereas $\times$ indicates that we use $L_1$ norm alternatively; $\sqrt{}$ in the column "pseudo-labels induced hypothesis space" means the hypothesis space for $f_2$ is created based on pseudo-labels of target samples, which is corresponding to our THS proposal in the main paper, where $\eta \neq 1$). We repeat the adaptation on VisDA dataset three times and record the average and the standard deviation of the accuracy.

| METHOD | OPTIMAL JOINT ERROR | CROSS MARGIN DISCREPANCY | PSEUDO-LABELS INDUCED HYPOTHESIS SPACE | VISDA |
|---|---|---|---|---|
| BASELINE | $\times$ | $\times$ | $\times$ | 71.9±0.4 |
| | $\times$ | $\sqrt{}$ | $\times$ | 73.1±0.3 |
| | $\times$ | $\times$ | $\sqrt{}$ | 76.2±0.3 |
| | $\times$ | $\sqrt{}$ | $\sqrt{}$ | 76.8±0.2 |
| OURS | $\sqrt{}$ | $\times$ | $\times$ | 73.4±0.3 |
| | $\sqrt{}$ | $\sqrt{}$ | $\times$ | 76.4±0.3 |
| | $\sqrt{}$ | $\times$ | $\sqrt{}$ | 79.2±0.2 |
| | $\sqrt{}$ | $\sqrt{}$ | $\sqrt{}$ | 81.6±0.2 |

study on VisDA dataset. The results in Tab. 8 show a similar conclusion that each part of our proposal is beneficial for adaptation, despite a more complicated scenario.

