# OpenReview forum: "Unsupervised Domain Adaptation via Minimized Joint Error"
_TMLR — Accepted by TMLR_

### Review · Reviewer_n4XW · 2023-04-04

**Summary Of Contributions:**

This paper tackles the problem of domain adaptation, where we assume access to a labeled source dataset as well as unlabeled data from the target dataset that we want to ultimately perform well on. Revisiting the bound (for the target error) proposed in early seminal work in the field by Ben-David et al, the authors remark that most recent work on domain adaptation ignores the ‘joint error’ term and focuses on learning embeddings where the discrepancy between the marginal distributions of the source and target is minimized. The issue that can arise from this is that the two distributions are indistinguishable from each other in terms of features, but with different classes mixed together in embedding space. Based on this observation, they propose an upper bound that takes the ‘joint error’ into account. Their upper bound is given in terms of the optimal labeling functions for the source and target, which in practice we don’t have access to, so they relax it to replace those by a supremum, introducing a min-max flavor into their optimization problem, reminiscent of other adversarial approaches. Then, they tighten the bound by constraining the space of hypotheses from which the supremum can be taken over: one hypothesis space is constrained to be the set of hypotheses that work on the source domain (this is easier to characterize as we have access to source labels) while the other is constrained to be the set of hypotheses at the intersection of the sets that work well on the source and target domains (for the latter, pseudolabels are used, obtained by the source model). They provide a practical algorithm for the resulting min-max optimization that has 3 phases, optimizing different sets of parameters with different sub-parts of the objective. They perform a thorough experimentation on different datasets showing that their approach sometimes outperforms some baselines.


**Audience:**

Yes

**Broader Impact Concerns:**

I have no broader impact concerns about this work in particular.

**Claims And Evidence:**

No

**Requested Changes:**

- [Related work] The authors say that the problem of ignoring the joint error ‘remains unsolved’, but several works mentioned in the related work section seem to address this, so it’s not clear which aspect is actually still unsolved. For instance, the authors effectively dismiss some previous work because they say that it involves hyperparameters. But don’t most methods have that property (including the proposed one)?

- [Related work] On a related note, why don’t the authors compare experimentally against the similarly-motivated methods mentioned in the related work: (Sener et al. 2016), (Zhang et al, 2018) and (Wu et al, 2019), for example?

- [Related work] On another related note, it would be great to elaborate more on the differences between the proposed method and this most related categories of works. The authors say they are the first to ‘directly’ minimize the joint error for UDA. Do other approaches minimize it indirectly then? If so, what are the benefits of the direct approach? Also, the authors say in the experiments that their approach is very similar to MCD. Please explain what exactly is the difference.

- [Clarity / Motivation] It wasn’t clear to me why the hypothesis space H2 is defined as the intersection the hypothesis space of source models and that of target models (according to pseudolabeling). Why not define it as just the latter, rather than the intersection of the two? On a related note, Figure 2 shows the ‘improper constraint’ in subfigure a (by the way, I’m not sure what ‘improper constraint’ vs ‘alleviated constraint’ means. Which constraint exactly is this referring to? Why is one improper?) where the improper constraint is setting the hypothesis space H2 to be that of the source models. But a more intuitive choice would be setting it to be that of the *target* models. It would be great to clarify why this is the reference point used there.

- [Clarity] A very prominent clarity issue is that the ‘joint error’ is at the forefront of the narrative of the paper (incl. the abstract, intro, etc) but it’s not actually defined (formally or intuitively) until very late (it’s defined the first time as far as I can tell only in page 3, *after* having presented the proposed upper bound even). It would really help readability to explain what exactly is meant by joint error.

- [Clarity] Confusing notation in several places. For example, the error \epsilon is defined as taking 2 arguments (corresponding to prediction and ground truth). The source error \epsilon_S should then be defined as \epsilon(h, f_S) rather than as \epsilon_S(h, f_S) – currently, \epsilon_S appears on the RHS of the equation that defines it.

- [Clarity] Furthermore, the definition of \epsilon_S says that it takes just 1 argument (the hypothesis h), because the second argument is fixed to be the optimal labeling function for the source. But then, several terms in Equation 1 use \epsilon_S (and correspondingly \epsilon_T for the target) with two arguments which is confusing.

- [Clarity] Also, in Equation 1, it is not clear what \epsilon_S(f_S, f_T) is versus \epsilon_T(f_S, f_T) – did the authors instead mean \epsilon(f_S, f_T) versus \epsilon(f_T, f_S)? Notice the different order of arguments there. I.e. these quantities are the error on the target induced by using the optimal hypothesis on the source, and the other way around (error on the source if using the hypothesis that is optimal for the target). Is this what was intended? If so, please correct. If not, can you please explain?

- [Clarity] In later equations, new symbols like \epsilon_{S_g} start appearing too, without being defined (as far as I can tell; might be missing it). Same for C_{S_g, T_g}. Same for f_{1or2}. I understand the meaning here for all of these, but it’s best to be precise, and introducing all these symbols that the reader must search (or infer) the definitions for causes fatigue.

- [Clarity] In Section 3.2.2 I find the text confusing. The authors talk about points being ‘outside’ of f2, for example. But f2 is a function. What does it mean to be outside a function?

- [Clarity] ‘All experiments are performed in an unsupervised fashion’ – this sentence confused me. I thought labels from the source domain were used e.g. in Step A of training?

- [Clarity] Add the missing ‘details’ regarding the pseudolabeling process. Are these computed online or offline? In stage 1 of training (Figure 4i), pseudolabels are used for the target loss that optimizes f2 (and g) – are the pseudolabels obtained from f1? These are learned jointly in this phase. At the start of training when f1 is random, is it possible that the produced pseudolabels are of very poor quality, which throws off the training of f2?

- [Experiment design] How are hyperparameters tuned?  \gamma and \eta seem to be particularly important (but also standard hyperparameters like learning rate, when to stop training, etc). This is important to discuss in UDA since we assume no labeled validation set from the target domain (by definition of the problem setting). So, it’s important to disclose how model selection happens

- [Experiment design] Related to the above, in some cases (e.g. in Table 2), the authors report several variants of their method (with different hyperparameters) and it seems that for different classes (columns in Table 2), different versions perform significantly better/worse than others. Ultimately when proposing a method though one needs to select a single model to deploy (we can’t decide based on the test set results which one to choose in each case…). So some of the results are showing that *there exists* a variant (i.e. setting of hyperparameters) that (sometimes) can perform well, without really offering a procedure for arriving at that variant.

- [Experiment design] Why do different tables compare against different subsets of baselines?

- [Soundness] The authors claim that their method sometimes outperforming others demonstrates that their proposed approach manages to penalize ‘the undesired matching between the source and target domain’. I don’t think this necessarily follows by simply looking at the numbers in the tables. Can’t there be several other reasons for the better or poorer performance of different models? It seems that the authors should design a separate experiment to investigate this in particular if they want to make this claim.

- [Soundness] The authors claim that one of their contributions is showing that their proposed novel measurement of dissimilarity can ‘alleviate the instability during adversarial learning’. But this isn’t formally defined (instability in terms of what?) and I don’t see where this is shown. There are some plots showing faster convergence using the proposed measurement compared to others, which might relate to but isn’t the same as mitigating ‘instability’.

- [Soundness] ‘Fig. 5b empirically proves simply minimizing the discrepancy between marginal distributions does not necessarily lead to a reliable adaptation’. This figure shows the marginal discrepancy over epochs, but I’m not sure how this relates to the quality of adaptation, and what evidence is used exactly to support this claim.


**Strengths And Weaknesses:**

Strengths
========
- The proposed approach is derived from a bound that is more appropriate than previous ones in that the joint error term is taken into account
- Thorough empirical investigation on the relevant datasets / benchmarks for domain adaptation
- In some cases, the proposed method outperforms others

Weaknesses
===========
- It’s not clear in practice how big a deal it is to ignore the joint error. While I agree that the scenario illustrated in Figure 1 is possible in principle, it’s not clear how often this actually occurs. It would thus be great to directly investigate whether the phenomenon illustrated in Figure 1 occurs in the baseline approaches used in the experiments, and whether (or how often) the ‘failure’ of those baselines (e.g. worse performance compared to the proposed method) is indeed associated with this phenomenon occurring.
- The paper has a number of clarity issues (see below)
- Some important aspects of experimental design are missing (from the main paper at least) that need to be discussed.
- The proposed method is in several cases outperformed by other baselines
- The proposed procedure seems complex in terms of stages of training (which involves several design choices to be made, e.g. which submodules are frozen/trainable in each phase) and has some key hyperparameters which isn’t clear how should be tuned
- Some issues with soundness of claims (see below)

---

> ### Author Response · Authors · 2023-04-15
> **Answers**
>
> We thank you for the valuable comments and we response to your concerns as follows:
> ***
> **As for the Weaknesses:**
>
> **Q: It's not clear in practice how big a deal it is to ignore the joint error.**
>
> A: The learning theory proposed by Ben-David shows the target error can be bonded by:
> $\epsilon_T(h) \leq \epsilon_S(h) + \sup_{f_1,f_2 \in H}|\epsilon_S(f_1,f_2)-\epsilon_T(f_1,f_2)| + \lambda $.
> If an adaptation method follows this learning theory or its extension, given a feature extractor and a discriminator with enough complexity and the technique like GAN, the source error and the marginal discrepancy can be easily minimized. Fig.7(c) in page 11 also shows that the marginal discrepancy can be actually aligned. However, none of the existing method can achieve a perfect adaptation especially when the domain gap is huge. This is because of the increasing joint error $\lambda$ while minimizing the other two terms. Initially before the alignment between domains, the joint error may be small. Since it is high likely that there exists a hypothesis to jointly classify both domains when they are far away. However, when merging the marginal distribution of two domains during learning procedure, if the joint error is not controlled, the mismatch of conditional distribution can not be properly penalized (Fig.1(b)). We present a theoretical approach that constructs an optimizable upper bound on the joint error for UDA. By minimizing the joint error, we significantly improve the adaptation performance when the domain gap is huge (Table 3., AR to CL, PR to CL, RW to CL). This experimental result proves our statement that the joint error is crucial for the adaptation especially when the domain gap is huge.
>
> As for the methods do not follow any learning theory, e.g. adaptation based on progressively pseudo labeling [1,2], it is hard to tell the reason of their failures as there is no learning theory to indicate how each loss of their objective is related to the generalized target error. As for potential reasons, they assume samples in the target domain are well clustered within the feature space and can be labeled by the euclidean distance from clusters obtained with K-means. The first assumption is not always true since the high dimensional features can lie in a low dimensional manifold (e.g., Swiss Roll dataset) where no cluster exists.  Besides, the labeling scheme can be vulnerable because K-means is sensitive to the initialization and the euclidean distance can suffer from the curse of dimensionality where there is little difference in the distances between different pairs of samples.
>
> After all, this work aims to propose a novel learning theory that can upper bound the joint error, which is complementary to the traditional learning theory. Investigation on potential failures of methods with no theoretical background is beyond the scope of this work.
>
> [1] Qian Wang and Toby P. Breckon. Unsupervised domain adaptation via structured prediction based selective pseudo-labeling. AAAI,  2020.
>
> [2] Hui Tang, Ke Chen Chen, and Kui Jia. Unsupervised domain adaptation via structurally regularized deep
> clustering. CVPR, 2020.
>
> **Q: The proposed method is in several cases outperformed by other baselines**
>
> A: SOTA performance is not the goal of this work. We present a theoretical approach that constructs an optimizable upper bound on the joint error for UDA to tackle the mismatch of conditional distributions when the domain gap is huge, which is complementary to traditional learning theory that ignores joint error. Several experimental results demonstrate our proposal can significantly improve the adaptation performance when the domain gap is huge.
>
> **Q: The proposed procedure seems complex in terms of stages of training  and has some key hyperparameters which isn’t clear how should be tuned**
>
> A: We add a table to describe the algorithm in page 9. Details related to hyper-parameter selection is described in A.7 page 22. In short, the hyper-parameters are set based on the validation performance on a separate dataset consisting of a few labeled target samples.

---

> ### Author Response · Authors · 2023-04-15
> **Answers**
>
> **As for Requested Changes:**
>
> **Q: It’s not clear which aspect is actually still unsolved.**
>
> A: Currently, a theoretical approach that could actually minimize the joint error has not been fully explored. E.g., [3] claims their proposal can also reduce the joint error along with the discrepancy between domains by progressively building a pseudo labeled target set. However, it reforms the joint error $\lambda$ as the sum of the error on the source and pseudo labeled target domain and a false label rate $\rho$ which can not be optimized:
> $\lambda = \min_h [\epsilon_S(h)+\epsilon_T(h)] \leq \min_h [\epsilon_S(h)+\epsilon_{T_l}(h)] + \rho$.
> From the equation above, given a pseudo labeled target domain $T_l$, the error term $[\epsilon_S(h)+\epsilon_{T_l}(h)]$ can be easily minimized. However, the false label rate $\rho$ is uncontrollable and can not be optimized, thus this approach does not necessarily reduce the joint error $\lambda$. Besides, the performance of this approach is directly related the accuracy of pseudo labels, which usually requires a complicated selecting procedure involving several hyper-parameters like threshold. On the contrary, we propose a valid upper bound for the joint error which can be optimized in practice regardless of the accuracy of pseudo labels (Showed in A.5, page 20).
>
> [3] Asymmetric Tri-training for Unsupervised Domain Adaptation. Kuniaki Saito, Yoshitaka Ushiku, Tatsuya Harada. ICML, 2017.
>
> **Q: Why don’t the authors compare experimentally against the similarly-motivated methods mentioned in the related work**
>
> A: Because all of the works the reviewer mentioned do not provide the experimental
> results of those difficult tasks like VisDA, Office-Home. Besides, beating the performance of a work published long ago does not make our paper persuasive.
>
> **Q: Do other approaches minimize it indirectly then?  Please explain what exactly is the difference from MCD.**
>
> A: We modify some statements that may be confusing.
> In this work, we present a theoretical approach that constructs an optimizable upper bound on the joint error for UDA.
>
> E.g., [4] claims that their approach can reduce the joint error by applying a stricter match between the marginal distributions through penalizing the eigenvectors with largest singular values in the feature representations, such that the other eigenvectors with relatively smaller singular values can be matched instead of being suppressed. However, it is not well explained in theory why this can reduce the joint error, which is a problem more related to the conditional discrepancy. We agree the algorithm can lead to a discriminative (well-clustered) feature space where the ratio of between-class variance to within-class variance is maximized, but it cannot guarantee the target samples are placed in the correct clusters. Once the target samples are placed in the wrong cluster, a larger joint error is inevitable.
>
> As for the difference from MCD:
>  - we propose a valid upper bound as well as the generalization error given by Rademacher Complexity based on solid theory; MCD follows Ben-David's learning theory that ignores the joint err and lacks solid theoretical analysis.
>  - we propose a new discrepancy measurement CMD based on margin theory and can be interoperated with CGAN, which can guarantee the alignment between conditional distributions; MCD measures the discrepancy with $L_1$ norm, which lacks theoretical background and violates the consistency with source error measured by cross entropy.
>
> [4] Transferability vs. Discriminability: Batch Spectral Penalization for Adversarial Domain Adaptation, ICML 2019

---

> ### Author Response · Authors · 2023-04-15
> **Answers**
>
> **Q: It wasn’t clear to me why the hypothesis space H2 is defined as the intersection**
>
> A: Sec. 3.2.2 has been revised.
> In this section, we try to explain the reason why we make such a constraint on $f_2$ in a more intuitive way. An improper hypothesis space constrain can cause wrong alignment. If $f_2$ is forced to only classify all source samples, it may completely misclassify all target samples especially when the domain shift is huge. Besides, $f_2$ is an approximation of the target labeling function $f_T$ which does not necessarily predict all source samples correctly. If $f_2$ can classify a part of source samples and pseudo labeled target samples, the feature extractor can align the distributions of source and target domain in a desired way by minimizing the upper bound (shadow area in Fig.2). As mentioned by the reviewer, if most of the pseudo labels are correct, we can set hypothesis space H2 to be that of the target models. However, it is not practical especially the domain gap is huge. From Fig.2(b), if $f_2$ can classify a part of source samples and pseudo labeled target samples, by minimizing our upper bound ($\epsilon_S(f_1,f_2),\epsilon_T(f_1,f_2)$), the feature space between domains can be aligned.
>
> **Q: It would really help readability to explain what exactly is meant by joint error.**
>
> A: We add an explanation about joint error in Sec.1 , Paragraph 2.
>
> **Q: Confusing notation in several places.**
>
> A: The notations used in this paper follows Ben-David's work:
> $\epsilon_S(h,f_S)=E_{x \in S}|h(x)-f_S(x)|$.
>
> $\epsilon_S(h)$ is short for $\epsilon_S(h,f_S)$.
>
> $\epsilon_S(f_S,f_T)$ indicates the disagreement between $f_S,f_T$ on $S$. The oder does not affect the quantity as it is a distance metric.
>
> $S_g,T_g$ are defined in Sec.3.2, paragraph 2. We forget to bold the text.
>
> We modify the expression to $f_1,f_2\in H_f$ and $f_{1}\circ g, f_{2}\circ g \in H$. It means we assign $f_1,f_2$ a new function space $H_f$ with less parameters compared to $H$.
>
> We hope these explanations can solve the concerns related to the notations.
>
> **Q: In Section 3.2.2 I find the text confusing. What does it mean to be outside a function?**
>
> A: We mean outside the decision boundary of $f_2$.
>
> **Q: All experiments are performed in an unsupervised fashion’ – this sentence confused me.**
>
> A: We mean the model parameters are learned without using target labels in any loss function. If this sentence is confusing, we can remove it.
>
> **Q: Add the missing ‘details’ regarding the pseudo labeling process.**
>
> A: The pseudo labels are given by predictions of $h$ in each iteration during online training (Eq.7, page 7; Alg.1, page 9). As mentioned by the reviewer, it is possible that the produced pseudo labels are of low quality initially. Therefore, we try to add a ramp-up function to ignore the pseudo labels at early stage. However, there is no big change in the performance.
>
> **Q: How are hyper-parameters tuned?**
>
> A: Details related to hyper-parameter selection is described in A.7 page 22. In short, the hyper-parameters are set based on the validation performance on a separate dataset consisting of a few labeled target samples. Learning rate is set according to [5]. Currently, there is no other reliable choice and without the validation set, most of the methods won't be able to provide a descent result including ours.
>
> [5] Bridging Theory and Algorithm for Domain Adaptation. Zhang, Yuchen & Liu, Tianle & Long, Mingsheng & Jordan, Michael. ICML,2019.
>
> **Q: the authors report several variants of their method**
>
> A: We provide several results in Table 2 to show how different hypothesis constraints can affect the adaptation, as well as to make a fair comparison with MCD (we deploy the same hypothesis constraint and discrepancy measurement to demonstrate the effectiveness of our upper bound). Our final result is given by $\gamma=\eta=0.9$ through all experimental settings.
>
> **Q: Why do different tables compare against different subsets of baselines?**
>
> A: Some works do not provide the results of all the benchmarks.

---

> ### Author Response · Authors · 2023-04-15
> **Answers**
>
> **Q: Can’t there be several other reasons for the better or poorer performance of different models?**
>
> A: As explained above, the learning theory proposed by Ben-David shows the target error can be bonded by the sum of source error, marginal discrepancy and joint error:
> $\epsilon_T(h) \leq \epsilon_S(h) + \sup_{f_1,f_2 \in H}|\epsilon_S(f_1,f_2)-\epsilon_T(f_1,f_2)| + \lambda $.
> Given the technique like GAN, the source error and the marginal discrepancy can be easily minimized. Fig.7(c) in page 11 also shows that the marginal discrepancy can be aligned in practice. However, none of the existing method can achieve a perfect adaptation especially when the domain gap is huge. When the marginal distributions of feature space are matched, the only reason that can harm the accuracy of the classifier trained on source is the mismatch of conditional distributions, which means the samples from different classes are merged across domains. This is caused by the increasing joint error $\lambda$ while minimizing the other two terms, since existing methods ignore $\lambda$ during the optimization. We present a theoretical approach that constructs an optimizable upper bound on the joint error for UDA. By minimizing the joint error, we significantly improve the adaptation performance when the domain gap is huge (Table 3., AR to CL, PR to CL, RW to CL). This experimental result proves our statement that the joint error is crucial for the adaptation especially when the domain gap is huge and our proposal is indeed effective.
>
>
> **Q: I don’t see where the definition related to instability on adversarial learning is shown.**
>
> A: The problem related to instability is explained in Sec.3.3, last paragraph. Fig.3(a) shows that our proposed loss CMD is more flat around the original which means it gives relatively small gradient when samples are close to the decision boundary. As showed in Fig.3(b), during the optimization of a minimax game, when two hypotheses try to maximize the discrepancy (striped area), if one moves too fast around the decision boundary such that the discrepancy is actually maximized w.r.t some samples, then these samples can be moved into either side to decrease the discrepancy, which may lead to a wrong alignment.
>
> **Q: I’m not sure how Fig.5(b) relates to the quality of adaptation**
>
> A: Fig.5(b) shows the change of marginal discrepancy during the learning of MNIST to SVHN. MCD following Ben-David's theory that ignores the joint error achieves a lower marginal discrepancy compared to ours. However, ours gives a far better adaptation performance in Table 1. This result shows that simply minimizing the discrepancy between marginal distributions does not necessarily lead to a reliable adaptation especially when the domain gap is huge.

---

> > ### Comment · Reviewer_n4XW · 2023-04-26
> > **response to authors**
> >
> > Thank you authors for your responses.
> >
> > RE: other baselines, I still think they should be compared against. They might have public code or otherwise reimplementing at least some of them would really strengthen the paper. I don't think it matters how 'old' they are (and anyway, 2019 isn't that long ago); if they are the most related group of baselines, it would be great to compare to see if the new method improves at least upon similarly-motivated ones (especially since it doesn't improve in general in several scenarios considered).
> >
> > RE: soundness of the claim that the experimental results prove that joint error is crucial: So, if I understand correctly, the authors' argument for this is that: 1) if the marginal distributions of the features are 'matched', the only reason for poor performance is the joint error and, in addition, 2) it's not too difficult to 'match' those marginals with current approaches. Therefore, the only reason for poor performance is joint error? This argument seems reasonable, though it would be great to quantify how matched those marginals are empirically. This would lead to indeed showing that the only remaining issue is the joint error, so any improvement in performance can be attributed to reducing that indeed.
> >
> > RE: "Our final result is given by gamma = eta = 0.9 through all experimental settings." - This particular setting is not even shown in Table 1. Why is that? In other cases (e.g. several cases in Table 4), this setting is outperformed by baselines.
> >
> > Some unaddressed notation issues; please read my original review. For example:
> > - 2 lines above of Equation 1, where the source error \epsilon_S is defined, it should be \epsilon(h, f_S) rather than as \epsilon_S(h, f_S) on the right hand side of that equation defining it
> > - Several terms in Equation 1 use \epsilon_S (and correspondingly \epsilon_T for the target) with two arguments, whereas these should take just one argument each, as per the definition (the subscript S and T already speak to the second argument)
> >
> >
> > Overall, a main concern that remains with the paper is that the proposed method seems quite complex and I'm not sure from the experimental results if this is really justified / worth it in practice. I do see that the authors point out that in situations with large domain shift, they outperform other methods e.g. some entries in Table 3, but in other parts of Table 3, the situation is reversed. So I'm not really sure what to make of this. Perhaps the authors should look into different benchmarks with more pronounced distribution shifts that better highlight the benefit of the proposed method? As is stands, the results seem quite mixed.  In addition, the authors don't compare against similarly-motivated (to take joint error into account) previous work, which perhaps would perform equally well on those very-out-of-distribution parts of Table 3, for instance. This makes it hard to place the proposed method into context.

---

> > > ### Author Response · Authors · 2023-04-27
> > > **Answers**
> > >
> > > We thank you for the valuable comments and we response to your concerns as follows:
> > > ***
> > > - We can add a comparison with the methods (e.g. [1]) that claim to reduce the joint error if it is crucial for the acceptance.
> > >
> > > - However reimplementing takes time and we can not guarantee to make the most of their algorithms that may requires a further fine-tune of the hyper-parameters for an untested benchmark.
> > >
> > > - Besides, we are not sure if it would be meaningful since [1] cannot even beat DANN [2] in a very simple task like USPS to MNIST
> > >
> > > [1]Domain adaptation with asymmetrically-relaxed distribution alignment. PMLR,2019
> > >
> > > [2]Domain-adversarial training of neural networks. JMLR,2016
> > >
> > > ***
> > > - Currently, there is no universal measurement that can quantify how the marginal distributions are matched.
> > >
> > > - The loss function of GAN can minimize the JS-divergence between distributions; however it requires to discriminator to take the optimal which is not practical.
> > >
> > > - Besides, the quantity of the loss itself usually does not reflect the similarity between distributions since we can alway increase the power of the generator to decrease the loss (the evaluation of GAN still heavily relies on human eyes).
> > >
> > > - We present the feature space visualized by t-SNE to show the alignment between domains, which is a common choice used by many other works. In, Fig.7(c), we show that the marginal distributions between domains are almost matched.
> > >
> > > ***
> > > - The last row in Tab.1 shows the result of $\gamma=\eta=0.9$ and we are sorry for the typos.
> > >
> > > - We do not think Office-31 (Tab.4) is an appropriate benchmark to evaluate DA methods since it only contains a few hundreds samples (including a lot of duplicates) for each domain.
> > >
> > > - We test on it only because many other works also make this evaluation.
> > >
> > > - Learning theory based methods like ours require an adequate sample size to bound the generalization error; while cluster based methods (e.g., [1,2]) usually prefer the benchmark like Office-31 but does not work on large dataset like VisDA.
> > >
> > > [1]Unsupervised domain adaptation via structured prediction based selective pseudo-labeling. AAAI,2020
> > >
> > > [2]Unsupervised domain adaptation via structurally regularized deep
> > > clustering. CVPR,2020
> > >
> > > ***
> > > It seems that there are still some misunderstandings about the notations:
> > >
> > > - For a binary classification problem, we define a distance metric $\epsilon_D(f_1,f_2)=E_{x \in D}|f_1(x)-f_2(x)|$.
> > >
> > > - $\epsilon_D(f_1,f_2)$ indicates the disagreement between $f_1,f_2$ on $D$.
> > >
> > > - The subscript $D$ represents a distribution that $\epsilon$ is measured on and it has nothing to do with the arguments inside $\epsilon$.
> > >
> > > - The order of $f_1,f_2$ also does not matter since $\epsilon$ is a distance metric.
> > >
> > > - When we want to refer to the source error of a hypothesis, we use the shorthand $\epsilon_S(h)=\epsilon_S(h,f_S)$. (Maybe this part cause the misunderstanding)
> > >
> > > - The second argument of $\epsilon$ is omitted only when we want to refer to the source (and target) error.
> > >
> > > - The definition of $\epsilon$ comes first and it takes 2 arguments; the above shorthand is not necessary and we use it just to follow the common rules.
> > >
> > > - All the notations including the shorthand are introduced since [1] and followed by plenty of works.
> > >
> > > [1]A theory of learning from different domains. Machine Learning, 2009

---

> > > ### Author Response · Authors · 2023-04-27
> > > **Answers**
> > >
> > > ***
> > > - Currently the SOTA performance of UDA benchmarks is really high and it would be very difficult to propose a method that can beat all the others in every aspect.
> > >
> > > - When domain shit is small, the joint error is likely to decrease along with the minimization of the marginal discrepancy. In such case, an upper bound of joint error would be meaningless while semi-supervised regularizations (e.g., [1,2] in Tab.3) can boost the performance. However, when domain shit is huge, an upper bound of the joint error becomes necessary as it may increase along with the minimization of the marginal discrepancy.
> > >
> > > - Those powerful semi-supervised regularizations (e.g., augmentation based pseudo labeling [3] and consistency loss [4]) can be easily plugged in our algorithm to boost the performance, but the goal of this work is not to win an accuracy contest.
> > >
> > > - This work aims to propose a novel learning theory that can upper bound the joint error, which is complementary to the traditional learning theory. Experimental evidence (e.g., some entries in Tab.3, ours improves the accuracy over 5%) shows our proposal is very effective to the joint error related problem and robust to the real world situations like imbalance label distribution (Tab.6, A.10, Page 25), which we think is already a remarkable contribution.
> > >
> > > - Our proposal provides a more reliable solution to the large domain shift situation, which we think is meaningful to real world problems. Since when we face a hard problem in practice, our method becomes the best choice.
> > >
> > > - We have already evaluated on popular benchmarks (5 datasets) that are frequently chosen by other works including 2 hard tasks (Office-Home, VisDA). E.g., [1] only evaluates on 3 datasets and two of them (Office-31, ImageCLEF) are easy tasks where their advantage is subtle as other works also achieve a high accuracy around 90%. Therefore, we think those results of hard scenarios (MNIST to SVHN; VisDA; AR to CL, PR to CL, RW to CL in Office-Home) can justify our proposal.
> > >
> > > - We achieve a better average score in 2 hard tasks VisDA (Tab.2) and Office-Home (Tab.3)
> > >
> > > - If an additional comparison can make the reviewer raise the score, we are willing to do it.
> > >
> > > [1]Unsupervised domain adaptation via structurally regularized deep clustering. CVPR,2020
> > >
> > > [2]Unsupervised domain adaptation via structured prediction based selective pseudo-labeling. AAAI,2020
> > >
> > > [3]Simplifying semi-supervised learning with consistency and confidence. NIPS, 2020.
> > >
> > > [4]Virtual adversarial training: a regularization method for supervised and semi-supervised learning, PAMI 2018

---

> > > ### Author Response · Authors · 2023-05-02
> > > **Additional Experiment**
> > >
> > > ***
> > > - We add a comparison in Tabel.3 with ADA [1] (reimplemented by ourselves), which claims to address the joint error problem.
> > > - Our proposal outperforms theirs in every scenario as well as the average score about 10%.
> > > - The potential reason is that their method only tackles label distribution shift while ours address large domain shift which includes label and feature distribution shift.
> > >
> > > [1]Domain adaptation with asymmetrically-relaxed distribution alignment. PMLR,2019

---

> > > > ### Comment · Reviewer_n4XW · 2023-05-02
> > > > **response to authors**
> > > >
> > > > Thank you for the responses. Please also clarify the notation in the paper, as you wrote in the response to me.
> > > > Thank you for the comparison with ADA and for the clarifications.
> > > > I see the argument that even if the proposed method helps on only a subset of tasks, this can still be very valuable for the community, especially if that subset is mostly disjoint from that in which previous methods work well, and large distribution shifts are certainly a very relevant problem.
> > > > Overall, the proposal to take joint error into account seems reasonable and the theory provided is a good contribution. Though I still find the paper hard to read, I will lean towards acceptance.

---

> > > > > ### Author Response · Authors · 2023-05-02
> > > > > **Thank You**
> > > > >
> > > > > - We clarify the notations based on the suggestions
> > > > > - We greatly appreciate your acknowledgement of our contributions to the community.

---

### Review · Reviewer_w1Z7 · 2023-04-06

**Summary Of Contributions:**

This paper proposes a novel unsupervised domain adaptation (UDA) method which takes into account the joint error (how much a classifier can be effective on both source and target at the same time). While in most traditional approaches to UDA ignore this term due to intractability, the paper proposes to minimize an upper bound of this joint error, which relies on pseudo-labels obtained on target samples. Authors propose a novel cross-margin discrepancy, and show results and empirical evidences of effectivity on classical UDA datasets (digits, VisDA, Office-Home, etc.).

**Audience:**

Yes

**Claims And Evidence:**

No

**Requested Changes:**

Overall, while I acknowledge that the authors made some good efforts to provide a clear and illustrated exposition of their ideas, I must admit I had troubles understanding the core concepts as there are some notation issues. I listed some below, but at this stage and from the current state of the paper, I need to make too many assumptions about the meaning of the theoretical concepts to really assert the quality of the paper, and the novleties wrt. state-of-the-art.


Minor comments:
 - on page 4, you write $H_f \leq H$. What does it mean ?
 - Section 3.2, you write $g \in H_g$. I am a little bit confused here, in wich sense is the feature extractor belonging to an hypothesis space ? Maybe the meaning of $H$ in your paper, though classical in the machine learning literature, should be more clearly specified ?
 - section 3.2.1, typo: 'the all the samples'
 - I had troubles understanding Figure 2 and the corresponding description in Section 3.2.2. I encourage the authors to work on this subsection.
 - Section 3.3. there is something unclear, if $f_1$ and $f_2$ are hypothesis function, with outputs are probability distributions over classes, how can we write $1-f_1$ or $\text{log}(f_1)$ ?

**Strengths And Weaknesses:**

Strengths:
- trying to minimize the Joint Error makes sense, and the proposed contribution is a reasonnable contribution in this direction
- Experimental results are good

Weaknesses
- there are some unclear parts in the derivation of the method and the notations. See my comment below.
- The claim that this paper is one of the first to try to 'directly minimize the joint error' (page 3) seems a little exagerated, and could be undertoned. Notably, I strongly encourage the authors to clearly present the differences with the work from Saito and colleagues [1]. Though this paper [1] does present a similar idea from a less theoretical perspective (though [1, Section 4] is enlighting), it seems to the reviewer that the overall approach is similar, and the differences with the proprosed CMD discrepancy should be more clearly stated. Howvere approaches, such as [2] or [3], should also be examined under the same lense.

[1] Asymmetric Tri-training for Unsupervised Domain Adaptation
Kuniaki Saito, Yoshitaka Ushiku, Tatsuya Harada Proceedings of the 34th International Conference on Machine Learning, PMLR 70:2988-2997, 2017.

[2] CyCADA: Cycle-Consistent Adversarial Domain Adaptation
Judy Hoffman, Eric Tzeng, Taesung Park, Jun-Yan Zhu, Phillip Isola, Kate Saenko, Alexei Efros, Trevor Darrell Proceedings of the 35th International Conference on Machine Learning, PMLR 80:1989-1998, 2018.

[3] Bridging Theory and Algorithm for Domain Adaptation. Zhang, Yuchen & Liu, Tianle & Long, Mingsheng & Jordan, Michael. (2019).

---

> ### Author Response · Authors · 2023-04-15
> **Answers**
>
> We thank you for the valuable comments and we response to your concerns as follows:
>
> ***
> **As for the Weaknesses:**
>
> - We modify some statements that may be confusing.
> In this work, we present a theoretical approach that constructs an optimizable upper bound on the joint error for UDA.
> - [1]claims their proposal can also reduce the joint error along with the discrepancy between domains by progressively building a pseudo labeled target set. However, it reforms the joint error $\lambda$ as the sum of the error on the source and pseudo labeled target domain $T_l$ and a false label rate $\rho$ which can not be optimized:
> $\lambda = \min_h [\epsilon_S(h)+\epsilon_T(h)] \leq \min_h [\epsilon_S(h)+\epsilon_{T_l}(h)] + \rho$
> From the equation above, given a pseudo labeled target domain $T_l$, the error term $[\epsilon_S(h)+\epsilon_{T_l}(h)]$ can be easily minimized. However, the false label rate $\rho$ is uncontrollable and can not be optimized, thus this approach does not necessarily reduce the joint error $\lambda$.
> On the contrary, we propose a valid upper bound for the joint error which can be optimized in practice. We add a table to describe the details of the proposed algorithm in page 9.
>
> - As for the difference from [1]:
>   - we propose a valid upper bound as well as the generalization error given by Rademacher Complexity based on solid theory; [1] lacks solid theoretical analysis and does not necessarily reduce the joint error.
>   - we propose a new discrepancy measurement CMD based on margin theory and can be interoperated with CGAN, which can guarantee the alignment between conditional distributions; [1] progressively creates a pseudo label target set and use it to train a target classifier, which does not involve any feature space alignment mechanism.
>
> - A detailed comparison with [3] is described in A.2.1, page 18:
>   - The main contribution of [3] is to bridge the gap between the theory and algorithm. During the derivation of the theory, the loss function $\epsilon$ is required to be a distance metric. However, most of works approximate the source error with cross entropy and sometimes measure the discrepancy using different methods like $L_1$ norm. [3] proposes a novel loss MDD that brides this gap. However, it still follows Ben-David's theory and builds the upper bound with the sum of the source error, marginal discrepancy and joint error which is ignored during the learning.
>
> - As for the comparison with [2]:
>   - [2] tackles UDA problem with image-to-image approach, which does not involve any theoretical analysis on joint error. This work is more related to [4,5]. Image-to-image translation approach may be effective when the domain gap is small such as the task for the city scene segmentation where the domain shift is usually causes by light condition. All the 3 woks [2,4,5] do not provide any experiment results on more difficult case like Office-Home, thus we doubt if this type of methods can actually work when the domain gap is huge.
>
> [4]M.Y. Liu and O. Tuzel. Coupled generative adversarial networks. NIPS, 2016.
>
> [5]M.Y. Liu, T. Breuel, J. Kautz. Unsupervised Image-to-Image Translation Networks. NIPS, 2017.
>
> ***
> **As for Requested Changes:**
>
> - We modify the expression to $f_1,f_2\in H_f$ and $f_{1}\circ g, f_{2}\circ g \in H$. It means we assign $f_1,f_2$ a new function space $H_f$ with less parameters compared to $H$.
>
>
> - $H$ here is a function space that contains a set of functions. Feature extractor $g$ is also a function, so we think it should lie in a specific function space $H_g$.
>
> - Sec. 3.2.2 has been revised.
> In this section, we try to explain the reason why we make such a constraint on $f_2$ in a more intuitive way. An improper hypothesis space constrain can cause wrong alignment. If $f_2$ is forced to only classify all source samples, it may completely misclassify all target samples especially when the domain shift is huge. Besides, $f_2$ is an approximation of the target labeling function $f_T$ which does not necessarily predict all source samples correctly. If $f_2$ can classify a part of source samples and pseudo labeled target samples, the feature extractor can align the distributions of source and target domain in a desired way by minimizing the upper bound (shadow area in Fig.2).
>
> - We modify some notations that may be confusing.
> $f(y|x)$ represents the prediction confidence on class $y$ given input $x$, which is a 1-dim value.

---

> > ### Comment · Reviewer_w1Z7 · 2023-05-04
> > **Paper still hard to read /decipher**
> >
> > Dear Authors,
> >
> > Thanks for your reply. I also read the discussion with the other reviewer. While at this stage I believe the paper contains interesting ideas and directions, I can not recommend it for acceptance due to clarity issues and rigor in the exposition of mathematical concepts.
> > I will give two examples:
> >  - Eq (3) defines a training objective, which a minimization problem under constraint. $H_1$ and $H_2$ are function spaces defined as the solution of another minimization problem (== argmin). What does it mean ? Is the solution of the argmin a singleton, is it not empty ? are the set $H_1$ and $H_2$ reduced to single elements ?
> >  - Regarding the notation $f(y|x)$: while I understand its meaning, it is not rigorous as this notation refers to probabilistic notations, but f is a function from R^d to the simplex of probability measures.
> >
> > These are just two examples. Unfortunately, I do not have the time nor it is my job to underline them exahustively. All in all, I really believe that the authors should undergo a major rewriting of all the theoretical aspects of the paper, by:
> >  - defining properly all the symbols used (e.g. H is a function space or an hypothesis space ?) and ensure consistency throughout the whole paper
> >  - avoid vague or confusing statements in the definitions.

---

> > > ### Author Response · Authors · 2023-05-04
> > > **Answers**
> > >
> > > We thank you for the valuable comments.
> > > - We submitted a major revision related to notations and Sec.3.1, 3.2, 3.3, 3.4 to make our algorithm more readable.
> > > - We response to your concerns as follows:
> > > ***
> > > - In Eq.3, $H_1=${$f_1|argmin_{f_1 \in H}[\epsilon_{S}(f_1)]$}.
> > > It means $H_1$ contains all hypotheses that can correctly recognize source samples. The solutions will be multiple on a finite set $S$ given a rich $H$.
> > > - $f$ gives a single value for binary classification and a vector of probabilities for multi-class classification.  Since this cause confusion, we unify the notations as:
> > > $f\in H: X\in R^D\to R^K$, whose output denotes the probability of the input to be categorized as each class.
> > > - $H$ is a hypothesis space while we think a hypothesis is also a function. But since our explanations cause misunderstandings, we remove notations like $H_g$ and introduce $g: X\in R^D\to R^F, h\in H^F: R^F\to R^K$

---

### Review · Reviewer_rDS2 · 2023-04-06

**Summary Of Contributions:**

This paper proposed an unsupervised domain adaptation method through minimized joint error. This is a resubmission.


**Audience:**

Yes

**Claims And Evidence:**

Yes

**Requested Changes:**

See my comments in Weakness.

**Strengths And Weaknesses:**

The version has better organization and clearer explanations such as equations. No significant changes in experiments.

Though this version addresses some of my concerns, I still feel confused about the training algorithm, e.g., Fig. 4. Is it possible to replace it with an algorithm listed in the paper (or along with the figure)? Also, I do not see a demo code for reproducing results, so I am suspicious that there may be some tricks in the implementation to achieve such performance. I strongly encourage the authors to release the code somewhere.

---

> ### Author Response · Authors · 2023-04-15
> **Answers**
>
> We thank you for the valuable comments and we response to your concerns as follows:
>
> - We add a table to describe the algorithm in page 9.
> - We provide a link to the demo code.

---

### Author Response · Authors · 2023-05-07
**Major Revision**

- We submitted a major revision related to notations and Section 3 to make our algorithm more readable.

---

### Decision · Action_Editors · 2023-05-23

**Recommendation:** Accept with minor revision

**Comment:**

This paper is a re-submission of an earlier TMLR paper.  One of the three reviewers was also a reviewer on the previous submission, and two of the reviewers are new.  The previous reviewer is happy with the current version and recommended acceptance.  One of the two reviewers was willing to recommend acceptance after the latest revision, and the final reviewer still had concerns about the clarity of the paper, though they did not see the final revision (their recommendation came in before the final revision).

I took a look at the revision.  For the most part, it seems that this revision addresses the concerns of the negative reviewer.  However, while reading the paper I do agree with the general concern amongst the new reviewers that there are some clarity issues here, and they continue to persist.  I think at this point it's mainly a polishing issue, as there are some lingering grammatical issues.  For instance, in the first paragraph of the introduction, the first sentence is not entirely grammatically correct, nor is the fourth.  There are many such places in the text like this.  I think this paper needs a full polish to fix these issues, and to ensure that the notation and clarity is OK.  With a bit more polish this paper should be good to go.

**Audience:**

Yes, this paper is certainly relevant to many in the TMLR audience.  Domain adaptation is a fundamental problem in the community.

**Claims And Evidence:**

Yes, the claims made in the paper are accurate and convincing.  In particular, the reviewers are quite happy with the experimental validation at this point.

---

> ### Author Response · Authors · 2023-05-27
> **Thank You**
>
> We sincerely appreciate all the hard work of the AE and reviewers.